# MAPS: Memory-Aware Predictive Scheduling Framework for Large Language Model Serving

Tiancheng Zhang [1]  Yulin Chen [2]  Yunfeng Zhao [1]  Shaoyuan Huang [1]  Cheng Zhang [3]  Xiaofei Wang [1]

## Abstract

The surge of large language model (LLM) applications on personal devices imposes massive, bursty workloads on cloud serving infrastructure. While prefill-decode disaggregation improves throughput and scalability, memory-bound decode instances often suffer from persistent load imbalance, as output lengths are unknown when requests arrive at the cloud. To address this, we propose MAPS, a Memory-Aware Predictive Scheduling framework tailored for disaggregated LLM serving. MAPS performs device-assisted speculative output length prediction overlapped with cloud-side prefilling, incurring negligible latency overhead. To handle generation uncertainty, MAPS applies uncertainty-aware calibration to derive output-length upper bounds with target coverage, enabling safe scheduling decisions. Building on these bounds, MAPS employs a hierarchical global-local scheduling strategy to mitigate inter-decoder queue buildup and intra-decoder head-of-line blocking. Extensive experiments on two real-world workloads and two LLMs show that MAPS significantly outperforms three state-of-the-art systems, reducing average end-to-end latency by 42.6% and tail latency by up to 84.8%.

## 1. Introduction

Large language models (LLMs) are increasingly deployed in interactive applications on mobile and personal devices (Wang et al., 2025a), spanning conversational assistants and question answering (Yan et al., 2025). To improve

[1]College of Intelligence and Computing, Tianjin University, Tianjin, China [2]The International Joint Institute of Tianjin University, Tianjin University, Fuzhou, China [3]Faculty of Digital Economics and Managements, Tianjin University of Finance and Economics, Tianjin, China. Correspondence to: Yunfeng Zhao <yfzhao97@tju.edu.cn>, Cheng Zhang <zhangcheng@tjufe.edu.cn>.

*Proceedings of the 43rd International Conference on Machine Learning*, Seoul, South Korea. PMLR 306, 2026. Copyright 2026 by the author(s).

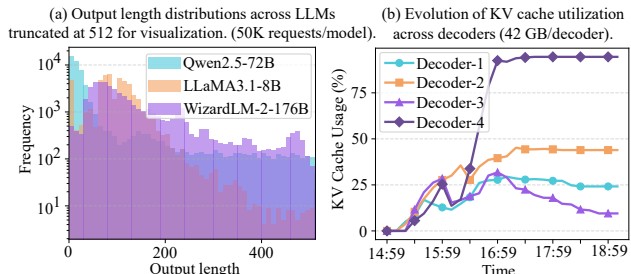

(a) Output length distributions across LLMs truncated at 512 for visualization. (50K requests/model).   (b) Evolution of KV cache utilization across decoders (42 GB/decoder).

*Figure 1.* Request and resource heterogeneity in LLM serving.

scalability and resource efficiency, modern LLM serving systems adopt prefill-decode (PD) disaggregation (Zhong et al., 2024), where prefilling is compute-bound while decoding is memory-bound due to the growth of KV caches with generated tokens. While scheduling at prefill instances is relatively straightforward given known input lengths, decode instances face unknown and highly variable output lengths that are progressively revealed during generation. As a result, existing systems rely on stateless routing policies such as round-robin (RR) or least-request (LR) routing, which avoid request-level demand information. Although LR appears more load-aware, it treats request count as a proxy for load and ignores decoding service time, making it vulnerable to saturation under long-generation requests. Consequently, state-of-the-art engines such as vLLM (Kwon et al., 2023) and SGLang (Zheng et al., 2024) continue to adopt the more conservative RR strategy.

However, the limitation of RR becomes increasingly pronounced as LLM output lengths grow in practice. As shown in Figure 1(a), real-world workloads exhibit highly skewed and heavy-tailed output length distributions (Aubakirova et al., 2026), leading to pronounced heterogeneity in decoding demand and uneven KV cache consumption over time. As a result, identically provisioned decoders can gradually diverge in KV cache utilization, with some becoming persistently saturated while others remain underutilized (Hu et al., 2024b), as illustrated in Figure 1(b) and further validated by queueing-theoretic analysis in Appendix A. Under such conditions, RR not only fails to balance load across decoders but can also exacerbate queue buildup and increase frequent preemption events (Sun et al., 2024). This problem is fur-

ther amplified by elastic deployments and heterogeneous decoder capacities in cloud clusters, motivating the need for more effective scheduling mechanisms.

To mitigate these limitations, prior work explores length-aware scheduling via output length estimation (Jin et al., 2023). PO-IT (Zheng et al., 2023) and Ranking (Fu et al., 2024) leverage length prediction or relative ordering to enable shortest-job-first (SJF) scheduling within a single decode instance. However, these methods remain confined to easily realizable intra-instance scheduling and do not extend to cross-instance request placement, as this would require additional runtime state monitoring, leaving long-generation requests susceptible to being dispatched to heavily loaded decoders. Moreover, prediction itself on the critical inference path introduces additional overhead, which can burden latency-sensitive LLM inference, and the inherent uncertainty in LLM generation further limits the efficiency of predictive scheduling. In contrast, works such as Llumnix (Sun et al., 2024) attempt to rebalance load through request migration after imbalance has occurred. While effective in some cases, post hoc migration introduces significant overhead for long-generation requests with large KV caches.

Taken together, enabling predictive scheduling in PD-disaggregated LLM serving entails three key challenges. 1) Output length prediction must be designed to incur minimal additional latency during inference. 2) Given the inherent uncertainty in LLM generation, prediction results must provide coverage guarantees to support low-risk scheduling decisions. 3) Effective scheduling must operate across both intra-instance and inter-instance levels to achieve low-latency inference under high-concurrency workloads.

To address these challenges, we propose MAPS, a Memory-Aware Predictive Scheduling framework for PD-disaggregated LLM serving. MAPS integrates a speculative prediction mechanism by finetuning a lightweight LLM deployed on request-origin devices, with prediction executed in parallel with cloud-side prefill to hide latency. To account for uncertainty in LLM generation, MAPS incorporates an uncertainty-aware calibration module at the cloud side, which derives coverage-guaranteed output length intervals based on historical prediction via conformal calibration. By bounding underestimation risk, these calibrated intervals prevent admitting requests without sufficient decoder memory, thereby avoiding excessive queueing and preemption.

Building on calibrated predictions, MAPS adopts a hierarchical global-local scheduling strategy. At the inter-instance level, MAPS performs memory-aware scheduling to balance decoder queue loads while ensuring sufficient resources based on predicted lengths, thereby reducing queueing delays and preemption risk. At the intra-instance level, each decoder applies SJF reordering to mitigate head-of-line (HOL) blocking and improve local execution efficiency.

Our main contributions are as follows:

- We design a device-assisted speculative prediction mechanism that overlaps output length estimation with cloud-side prefill, enabling early visibility into request-level resource demand with negligible latency overhead.

- Based on conformal calibration, we develop an uncertainty-aware calibration module that converts raw length estimates into reliable bounds, mitigating underestimation risk to ensure safe admission and memory feasibility for downstream scheduling decisions.

- Building on the calibrated bounds, a hierarchical global-local scheduling strategy is introduced to jointly mitigate inter-decoder queue buildup and intra-decoder head-of-line blocking, thereby improving inference latency and tail stability under high concurrency.

Across multiple LLMs and workloads, MAPS reduces average end-to-end latency by 42.6% and tail latency by up to 84.8% compared to three baselines.

## 2. Related Work

**LLM Serving Systems.** Recent LLM serving systems focus on improving inference efficiency at scale, with an emphasis on low latency under high concurrency. At the system level, SpotServe (Miao et al., 2024), FastServe (Wu et al., 2026), and AlpaServe (Li et al., 2023) explore cost-efficient execution, preemptive scheduling, and pipeline parallelism to improve latency under bursty workloads, while Dist-Serve (Zhong et al., 2024) adopts PD disaggregation to mitigate phase interference under strict latency constraints.

Prior systems improve inference efficiency primarily by optimizing execution within a single instance. Orca (Yu et al., 2022) introduces continuous batching, DeepSpeed-MII (Microsoft, 2023) enables large-model execution through model parallelism, vLLM (Kwon et al., 2023) improves memory efficiency with PagedAttention, and SGLang (Zheng et al., 2024) reduces redundant prefill computation via RadixAttention. However, in multi-instance deployments, these systems rely on reactive scheduling policies (e.g., RR), as the final generation length is unknown at dispatch time. Under heterogeneous workloads, such policies can lead to memory contention and queueing delays. Llumnix (Sun et al., 2024) partially alleviates this issue via runtime rescheduling and request migration, but remains reactive rather than predictive.

**Length-Aware Scheduling.** Early output-length prediction methods focus on non-autoregressive generation tasks (Gu & Tan, 2022; Shu et al., 2020), such as machine translation where output length strongly correlates with input length,

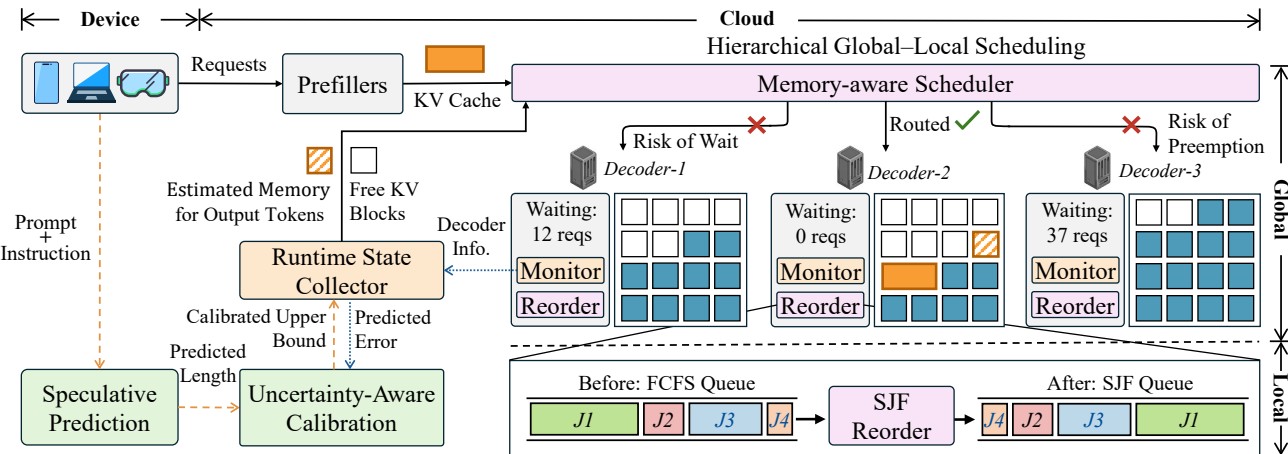

*Figure 2.* Overview of MAPS. MAPS consists of three paths: (i) the inference path (black), where requests are processed by the prefiller and decoders; (ii) the prediction path (orange), where speculative length prediction and uncertainty-aware calibration estimate future memory demand; and (iii) the state feedback path (blue), where runtime states are collected to guide scheduling decisions.

and do not directly generalize to autoregressive LLM inference with highly variable and uncertain generation lengths.

Recent studies explore predicting output length for LLMs to improve serving efficiency (Jin et al., 2023; Hu et al., 2024a; Stojkovic et al., 2025). Some approaches rely on dataset-level statistics or lightweight learned predictors (Qiu et al., 2024b;a; Sanh et al., 2019; Zhang et al., 2022; Cheng et al., 2024), often casting length prediction as a classification task. PO-IT (Zheng et al., 2023) prompts LLMs to self-estimate output length, while ranking (Fu et al., 2024) approximates SJF through relative request ordering without explicit length prediction. However, these methods implicitly assume that predicted lengths or orderings reliably reflect request-level resource demand, an assumption that frequently breaks under real-world workloads with high length uncertainty.

**Collaborative LLM Inference Across Device and Cloud.**
Recent advances in hardware capabilities and model compression techniques have made it increasingly practical to run small or quantized LLMs on end devices (Lin et al., 2024), spurring growing interest in collaborative LLM inference across device and cloud. Existing work in this area primarily focuses on execution-centric cooperation, including splitting LLM execution between devices and the cloud to improve throughput (Mudvari et al., 2024; Yang et al., 2024), as well as adaptive execution partitioning under dynamic network conditions (Younesi et al., 2025). In contrast, little attention has been paid to leveraging device-side inference for predictive scheduling decisions to assist cloud-side LLM serving, which is the focus of this work.

## 3. MAPS Design

MAPS leverages the complementary roles of device and cloud to construct three coordinated paths for memory-

aware predictive scheduling, as illustrated in Figure 2. Upon submission, each request follows two parallel paths. Along the inference path, the request is forwarded to the cloud-side prefiller, where prompt prefilling is performed. In parallel, along the prediction path, MAPS executes speculative prediction directly on the request-origin device without additional data transfer. The predicted length is then sent to the cloud-side uncertainty-aware calibration module, which refines it into a calibrated upper bound with statistical coverage guarantees and stores the result in the runtime state collector. After prefilling completes, the memory-aware scheduler retrieves the calibrated bound and jointly considers the predicted memory demand and the available KV cache capacity across all decoders. The request is then routed to a feasible decoder with sufficient free KV blocks and a short waiting queue. Within the selected decoder, MAPS applies SJF reordering among admitted requests. Basic mechanisms such as continuous batching and KV cache management are handled by the underlying vLLM runtime.

### 3.1. Device-Assisted Speculative Prediction

To enable proactive memory-aware scheduling, MAPS requires an early estimate of each request's output length prior to decoding. This estimation must balance accuracy and efficiency: inaccurate predictions degrade downstream scheduling decisions, while high-latency inference directly inflates time-to-first-token (TTFT).

An LLM is a natural choice for output length prediction (Zheng et al., 2023), as generation length depends not only on surface prompt statistics but also on higher-level semantic intent in the input. LLMs are explicitly trained to capture such semantic intent, which is difficult to model using simple statistical or regression-based predictors. However, directly invoking LLMs for prediction incurs addi-

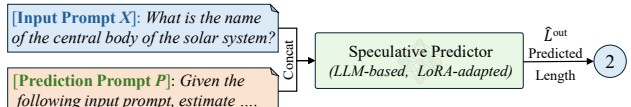

*Figure 3.* Device-assisted speculative prediction in MAPS.

tional latency, making them impractical for online serving. MAPS therefore adopts a device-assisted speculative predictor (SP), in which a lightweight LLM executed on the request-origin device produces early length estimates that overlap with cloud-side prefill, striking a practical balance between prediction fidelity and latency.

As shown in Figure 3, SP takes the input prompt $X$ together with a prediction prompt $P$ and, after a prefill pass, outputs a scalar $\hat{L}_{\text{out}}$ estimating the number of tokens to be generated.

$$\hat{L}^{\text{out}} = f_{\text{pred}}([X; P]). \tag{1}$$

Rather than predicting its own response length, the predictor is instructed to estimate the maximum output length that LLMs would generate for the input, improving robustness across heterogeneous models and decoding behaviors. The prediction prompt $P$ is shown below:

> Given the following input, estimate the max number of output tokens that LLMs would generate when responding to it. Return a single non-negative integer only.

However, prompt-based inference alone is insufficient to robustly capture generation variability across heterogeneous downstream models. MAPS therefore adapts the SP via parameter-efficient fine-tuning, using LoRA (Hu et al., 2022) with supervision aggregated from multiple LLMs. Specifically, each training target is defined as the maximum output length observed across LLaMA and Qwen models under multiple temperatures, encouraging the predictor to learn a conservative upper bound suitable for safe memory-aware scheduling. Training details are provided in Appendix B.

In our current implementation, MAPS employs a unified SP to estimate output lengths across downstream models. More generally, SP can be instantiated using lightweight variants aligned with the serving model, preserving compatibility with cloud-side inference and enabling efficient deployment.

### 3.2. Uncertainty-Aware Calibration

Although SP is trained to produce conservative maximum estimates, LLM generation remains inherently stochastic under highly non-stationary real-world workloads. Variability from user behavior, decoding temperatures, and workload shifts makes output length difficult to predict reliably. In production systems, underestimation can trigger queue buildup and preemption leading to multi-fold latency inflation (Kwon et al., 2023). As a result, prediction accuracy alone is insufficient; what matters for memory-aware scheduling is the reliability of predictions under uncertainty.

To make length prediction usable for online serving, MAPS incorporates an uncertainty-aware calibration module (UAC) that accounts for generation uncertainty by calibrating predictions based on recent runtime prediction errors. Specifically, MAPS adopts conformal prediction (Angelopoulos & Bates, 2023), a distribution-free framework that transforms point estimates into uncertainty-aware length intervals suitable for memory budgeting and scheduling. Conformal prediction requires no distributional assumptions and provides finite-sample guarantees, making it well-suited for heterogeneous LLM workloads. To adapt to distribution shift in non-stationary serving environments, MAPS maintains a sliding calibration window over recent requests and empirically achieves near-target coverage in practice.

We maintain a calibration set $\mathcal{D}_{\text{cal}} = \{(X_i, L_i^{\text{out}}, \hat{L}_i^{\text{out}})\}_{i=1}^n$ of the $n$ most recent completed requests, where $L_i^{\text{out}}$ denotes the observed generation length and $\hat{L}_i^{\text{out}}$ is the corresponding prediction. For each sample, we define a one-sided nonconformity score that captures underestimation error:

$$s_i = \left(L_i^{\text{out}} - \hat{L}_i^{\text{out}}\right)_+ = \max\left(0, L_i^{\text{out}} - \hat{L}_i^{\text{out}}\right), \tag{2}$$

which is positive only when the predictor underestimates the true output length. Given a target miscoverage rate $\alpha \in (0, 1)$, we compute the empirical $(1 - \alpha)$-quantile of the one-sided nonconformity scores:

$$q_\alpha = \text{Quantile}\left(\{s_1, \ldots, s_n\}, \lceil(1 - \alpha)(n + 1)\rceil\right). \tag{3}$$

Given a speculative prediction $\hat{L}^{\text{out}}$, the calibrated upper bound $\hat{L}^{\text{up}}$ on the output length is defined as:

$$\hat{L}^{\text{up}} = \hat{L}^{\text{out}} + q_\alpha. \tag{4}$$

Under the exchangeability assumption (Vovk et al., 2005), the calibrated upper bound satisfies the following guarantee:

$$\mathbb{P}\left(L^{\text{out}} \leq \hat{L}^{\text{up}}\right) \geq 1 - \alpha, \tag{5}$$

ensuring an underestimation rate no greater than $\alpha$.

### 3.3. Hierarchical Global-Local Scheduling

Our scheduling focuses on the memory-bound decoding phase. We model the decoding cluster as a set of decoders $\mathcal{D} = \{d_1, \ldots, d_K\}$, where each decoder $d_k$ has a KV cache capacity $C_k$. Leveraging PagedAttention, this capacity is virtualized into fixed-size blocks of size $S$, corresponding to $S$ tokens each, and the available resource of $d_k$ is measured by its number of free blocks. For each request $i$, we define a job tuple $J_i = (t_i^{\text{arr}}, L_i^{\text{prom}})$, where $t_i^{\text{arr}}$ is the arrival time and $L_i^{\text{prom}}$ denotes the prompt length.

**Runtime State Collection (RSC).** MAPS maintains a lightweight RSC to support scheduling decisions. Each decoder $d_k$ embeds a local monitor that periodically reports its runtime state, including the number of free blocks $B_k^{\text{free}}$, reserved blocks $B_k^{\text{res}}$ allocated to ongoing requests, effective blocks $B_k^{\text{eff}} = B_k^{\text{free}} - B_k^{\text{res}}$ available for scheduling, and the current queue length $Q_k$. Together, these signals capture real-time memory availability and pressure per decoder. For each incoming request $i$, the RSC receives the calibrated upper bound $\hat{L}_i^{\text{up}}$ and maintains a sliding window of recent request outcomes to support online calibration.

**Memory-Aware Scheduler.** Upon the arrival of a request, MAPS first retrieves the calibrated upper bound $\hat{L}_i^{\text{up}}$ from UAC. The scheduler then extends the job tuple to $J_i = \left(t_i^{\text{arr}}, L_i^{\text{prom}}, \hat{L}_i^{\text{up}}\right)$. If the upper bound is not yet available, the scheduler waits up to a predefined timeout $\beta$. Upon timeout, it falls back to RR dispatch to avoid stalling the inference pipeline. Once $J_i$ is constructed, MAPS estimates the KV cache demand of the request as:

$$B_i^{\text{need}} = \left\lceil \frac{L_i^{\text{prom}} + \hat{L}_i^{\text{up}}}{S} \right\rceil. \qquad (6)$$

A decoder $d_k$ is deemed feasible for job $J_i$ if its $B_k^{\text{eff}}$ can accommodate the $B_i^{\text{need}}$, as defined in Algorithm 1 (Appendix C). This feasibility check ensures that requests are preferentially dispatched to decoders with sufficient memory capacity, preventing overcommitment and preemption during decoding. Among all feasible decoders $\mathcal{D}_{\text{feasible}}$, MAPS selects the one with the minimum waiting queue length $Q_k$. This design directly targets queueing delay, which is the dominant contributor to tail latency under high load.

While routing based on maximum available capacity may appear intuitive from a memory-balancing perspective, it ignores queueing dynamics and can concentrate bursty requests on a single decoder with temporarily abundant memory, increasing waiting time and KV cache transmission overhead. This behavior is observed in our ablation study. Only in the worst case where no decoder satisfies the feasibility condition, MAPS falls back to this policy. This fallback strategy prioritizes memory availability to maximize the chance of forward progress. Once a target decoder is selected, MAPS atomically reserves $B_i^{\text{need}}$ to avoid race conditions under concurrent arrivals, ensuring that the memory feasibility condition is preserved during execution.

**Local SJF Reordering.** Once a request is routed to $d_k$, it is enqueued for execution, where local scheduling focuses on temporal efficiency by minimizing average latency and mitigating HOL blocking caused by long-running requests.

Traditional first-come-first-served (FCFS) policy is suboptimal for workloads with heterogeneous output lengths, especially under high load, as short requests may wait behind long-running ones (Fu et al., 2024). While SJF scheduling is known to minimize average waiting time in many systems (Harchol-Balter, 2013), it is rarely adopted in LLM serving. This gap arises because existing systems are largely agnostic to the execution time of incoming requests.

Under a fixed model architecture, decoding proceeds token by token with stable per-token latency. As a result, the total decoding time is approximately proportional to the output length, making the predicted length $\hat{L}^{\text{up}}$ a practical proxy for execution duration $T(J_i)$ in scheduling decisions, i.e., $T(J_i) \propto \hat{L}^{\text{up}}$. Each decoder maintains a ready queue $\mathcal{Q}_k$, which is dynamically reordered according to $\hat{L}^{\text{up}}$:

$$\text{Order}(\mathcal{Q}_k) = \text{argsort}_{J_i \in \mathcal{Q}_k}\left(\hat{L}_i^{\text{up}}\right). \qquad (7)$$

To mitigate starvation, MAPS enforces a maximum waiting time constraint that promotes requests exceeding a predefined threshold. Its sensitivity is analyzed in Appendix E. By prioritizing jobs with smaller $\hat{L}^{\text{up}}$ while bounding waiting time, the local scheduler reduces average completion time without starving long-generation requests.

### 3.4. Timeline Analysis

To illustrate where the latency gains of MAPS originate, we analyze the end-to-end inference timeline in Appendix D. By overlapping device-side prediction and calibration with cloud-side prefilling, MAPS effectively hides the prediction overhead from the critical path. Although MAPS introduces an additional memory-aware scheduling step, its overhead is negligible compared to decoding latency. The dominant latency reduction occurs during the decoding stage, where hierarchical global-local scheduling significantly reduces queueing delays and HOL blocking across decoders.

## 4. Experiments

**Implementation Details.** We implement MAPS on top of the vLLM inference engine (v0.13.0) and conduct all experiments on a cloud deployment with six NVIDIA A6000 GPUs in a PD-disaggregated setup, consisting of one prefiller and two decoders. KV cache transfer between the prefiller and decoders is enabled via Mooncake. On the device side, SP is hosted on NVIDIA Jetson Orin NX devices. To account for the heterogeneous resource budgets of edge devices, we deploy two SP variants of different scales: SP-7B, which uses Vicuna-7B as its backbone and reflects the growing on-device capabilities of modern edge hardware, and SP-160M, a lightweight variant built on Vicuna-160M that targets devices with more constrained memory. Unless otherwise specified, subsequent experiments use SP-7B as the default predictor, and MAPS refers to the system instantiated with SP-7B.

UAC performs conformal calibration with $\alpha = 0.1$ over

a sliding window of the most recent 500 requests. MAPS employs a lightweight runtime state collector implemented with Redis (v7.1.0) to aggregate decoder runtime states and prediction signals. The maximum waiting time and predefined timeout $\beta$ are set to one second.

As discussed in the Introduction, long-running LLM serving naturally induces load divergence across decoders due to heterogeneous request lengths and KV cache accumulation. To isolate and systematically study this effect, we intentionally configure different effective GPU memory utilization caps across instances, creating controlled memory heterogeneity in the serving environment. Specifically, for LLaMA3.1-8B, the prefiller and one decoder use a utilization cap of 0.8, while the other decoder is limited to 0.5; for Qwen2.5-3B, the corresponding caps are 0.6 and 0.2. Unless otherwise specified, all remaining system parameters follow the default vLLM configurations.

**Baselines.** For prediction, we compare against two representative approaches: PO-IT (Zheng et al., 2023), which adopts an LLM-assisted perception-only strategy via instruction tuning, and S3 (Jin et al., 2023), which formulates length prediction as a classification problem over discretized intervals. For scheduling performance, we compare MAPS with three widely used open-source inference frameworks: vLLM (v0.13.0) (Kwon et al., 2023), SGLang (v0.5.5) (Zheng et al., 2024), and Llumnix (Sun et al., 2024). Both vLLM and SGLang are evaluated using their official Mooncake-based PD-disaggregated deployments, while Llumnix follows its default Ray-based PD configuration.

**Workloads & Models.** We evaluate MAPS on two real-world workloads. Following prior work (Kwon et al., 2023), we construct synthetic workloads from ShareGPT (RyokoAI, 2023) and model request arrivals as a Poisson process with configurable rates. To capture realistic bursty behavior, we additionally evaluate on the Burst-GPT (Wang et al., 2025b) trace, which records bursty production request arrivals from Azure OpenAI and is replayed with temporal scaling to vary load intensity. Experiments are conducted on two widely adopted LLMs, LLaMA3.1-8B (Touvron et al., 2023) and Qwen2.5-3B (Qwen Team, 2024). These models differ substantially in size and decoding characteristics, enabling us to evaluate the robustness of MAPS across heterogeneous LLM backends.

*Table 1.* Prediction accuracy on the Alpaca dataset. Acc-$k$ denotes the percentage of requests with prediction error within $k$ tokens.

| Method | Backbone | MAE ↓ | Acc-50 ↑ | Acc-100 ↑ |
|--------|----------|-------|----------|-----------|
| **SP-7B** | Vicuna-7B | **56** | **58%** | **85%** |
| SP-160M | Vicuna-160M | 92.2 | 45% | 65% |
| PO-IT | Vicuna-7B | 63 | 56% | 81% |
| S3 | DistilBERT | 71 | 48% | 64% |

## 4.1. Output-Length Prediction Performance

**Point Prediction Accuracy.** This section evaluates the prediction accuracy of SP. For fair comparison, we follow the setting of PO-IT and conduct experiments on 20K randomly sampled prompts from the Alpaca dataset (Taori et al., 2023). The first 10K prompts are used for fine-tuning, while the remaining are for evaluation. The same data split is applied to both SP-7B and SP-160M. For S3, we follow its original setup and fine-tune DistilBERT on the same 10K prompts. Since S3 formulates length prediction as a 10-bucket classification problem, we use the upper bound of the predicted bucket as its length estimate for a conservative comparison.

Table 1 compares SP with existing prediction methods. In terms of mean absolute error (MAE), SP-7B reduces error by 11.1% and 21.1% over PO-IT and S3, respectively, indicating more accurate point estimates. Following PO-IT, we further report Acc-50 and Acc-100 to assess prediction reliability, where SP-7B consistently achieves higher accuracy. The lightweight SP-160M, as expected, yields higher MAE and lower Acc-$k$ due to its substantially smaller capacity. Nevertheless, its predictions still provide useful signals for downstream scheduling and are further refined by UAC, allowing MAPS to remain effective even under a tightly constrained predictor budget, as shown later in Section 4.3. Overall, these results show that SP improves both average accuracy and reliability within practical error tolerances. Additional error distribution analyses are provided in Appendix B.

*Table 2.* Underestimation risk analysis for output length prediction.

| Setting | Underestimation Prob. ↓ | Underestimation Err. ↓ |
|---------|-------------------------|-------------------------|
| SP | 64% | 503 |
| SP + Offset | 56% | 501 |
| **SP + UAC** | **9%** | **377** |

**Calibration for Safer Scheduling.** In output length prediction for LLM serving, prediction errors are inevitable due to inherent uncertainty. Among them, underestimation is far more harmful than overestimation, as it may trigger queue buildup, preemption, and severe tail-latency inflation. In contrast, moderate overestimation mainly incurs conservative resource allocation, whose impact is further mitigated by techniques such as PagedAttention. We therefore focus on underestimation-related risks in this section.

To quantify this risk, we report both the probability of underestimation and the maximum underestimation error. As shown in Table 2, the uncalibrated predictor exhibits a non-negligible underestimation probability, with severe errors on a subset of requests. A simple one-sided offset heuristic (SP+Offset), which inflates predictions by a fixed 10%, only marginally reduces the underestimation probability and fails to effectively bound the severity of underestimation. In

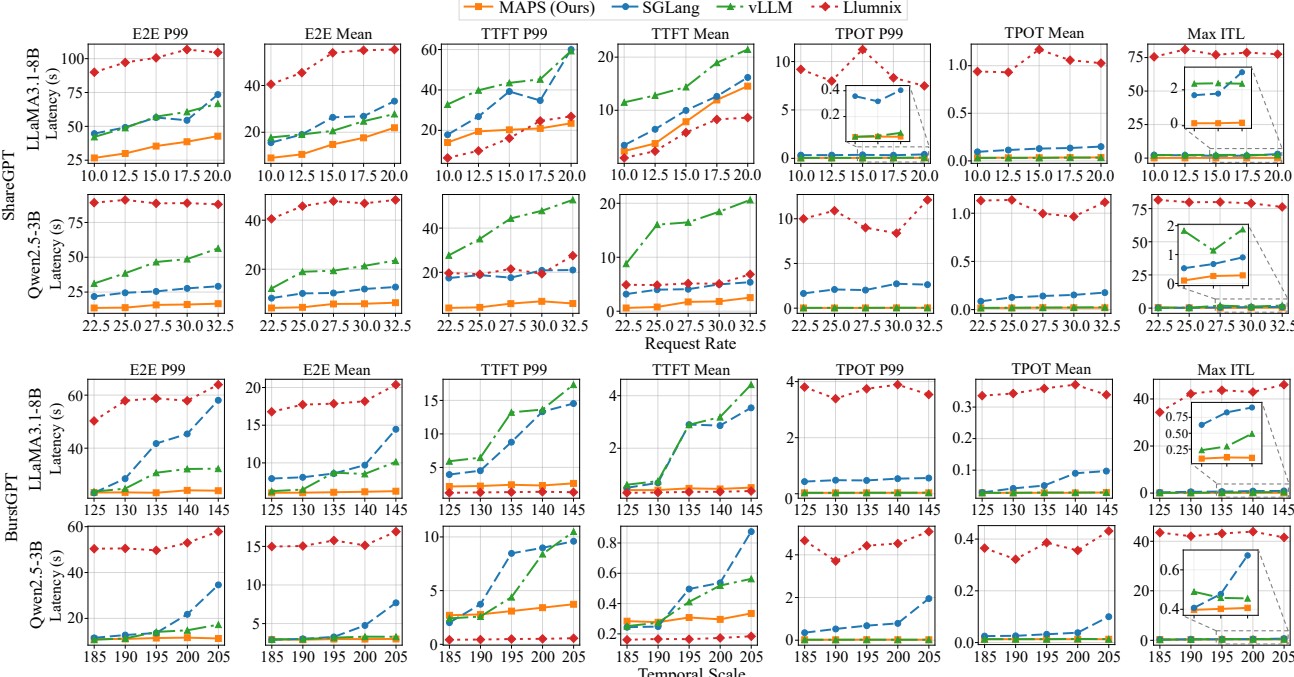

*Figure 4.* Scheduling performance on LLaMA3.1-8B and Qwen2.5-3B under different workloads.

contrast, applying UAC (SP+UAC) substantially lowers the underestimation probability and effectively caps residual errors via calibrated upper bounds. Even when underestimation still occurs, its impact is bounded, causing the system to degrade to behavior comparable to stateless dispatching.

## 4.2. Scheduling Performance

As shown in Figure 4, scheduling performance is primarily evaluated using end-to-end latency (E2E), measuring the total time from request submission to completion and directly reflecting service-level responsiveness. We additionally report two fine-grained latency metrics: TTFT, defined as the latency from request arrival to the first generated token, and time-per-output-token (TPOT), the average decoding latency per token. We also report the maximum inter-token latency (max ITL) to capture worst-case decoding interruptions under bursty workloads. Following the default configuration, MAPS in this section uses SP-7B as its predictor.

**End-to-End Latency.** Under ShareGPT workloads, MAPS consistently achieves the lowest P99 and mean E2E latency across all request rates. Although E2E latency increases for all systems as load grows, the baselines exhibit substantially steeper tail inflation, particularly in P99, whereas MAPS maintains stable latency. Compared with the three baselines, MAPS reduces P99 E2E latency by up to 70.3% across multiple request rates, indicating effective suppression of queue buildup and tail amplification under high load.

The advantage of MAPS becomes more pronounced under the BurstGPT workload, which features highly bursty arrivals. As the temporal scale increases, all baselines suffer from severe tail-latency inflation, with P99 E2E latency rising sharply due to escalating queueing delays. In contrast, MAPS consistently bounds P99 E2E latency and achieves an average reduction of up to 36.8% compared to all baselines. These results demonstrate that prediction-guided, hierarchical global-local scheduling is critical for mitigating worst-case request delays under highly bursty workloads.

Aggregated across both workloads and all evaluated load levels, MAPS reduces mean E2E latency by 42.6% and tail (P99) E2E latency by up to 84.8% over existing systems.

**TTFT and TPOT Analysis.** We further analyze TTFT and TPOT to understand the sources of E2E latency improvements. Under ShareGPT workloads, MAPS consistently achieves the lowest P99 TTFT at moderate and high request rates, even though its mean TTFT is slightly higher than Llumnix. This indicates that MAPS effectively controls tail admission latency by preventing requests from being queued behind memory-constrained decoders.

In contrast, Llumnix prioritizes dispatching requests to the most idle decoders, which can reduce average TTFT. However, as decoding progresses and output lengths grow, KV cache consumption may gradually exceed the available memory budget on the assigned decoder. Such memory pressure forces the system to either block decoding or trigger request migration, incurring substantial overhead. These

effects manifest as pronounced tail amplification, reflected in elevated P99 TTFT and extremely large TPOT. Meanwhile, SGLang and vLLM exhibit the opposite behavior: their TTFT grows rapidly relative to TPOT under high load, as stateless RR routing may admit requests to already saturated decoders, causing prolonged waiting before decoding can begin. MAPS also exhibits advantages in TPOT, especially in tail behavior. While mean TPOT is comparable across systems, the P99 TPOT of the baselines grows as load intensifies, indicating unstable decoding and frequent token-level stalls.

**Maximum ITL.** Llumnix consistently exhibits the largest max ITL across workloads and models, consistent with its reliance on request migration under memory pressure. Once migration is triggered, particularly for requests that have already generated most of their output, the interruption inevitably leads to extreme token-level delays. SGLang and vLLM also exhibit elevated max ITL. Although they do not rely on explicit migration, their stateless RR routing can dispatch requests to already heavily loaded decoders, leading to contention for KV cache and decoding slots. In contrast, MAPS maintains consistently low and stable max ITL across all settings, indicating that severe token-level interruptions are largely avoided. This highlights the efficiency of prediction-guided, memory-aware scheduling in stabilizing token-level progress under high load.

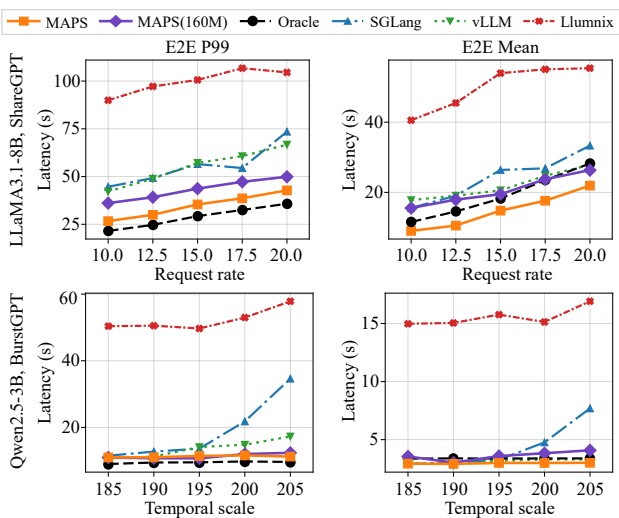

*Figure 5.* Robustness of MAPS under different prediction quality.

### 4.3. Robustness to Prediction Quality

As shown in Figure 5, we evaluate MAPS under two contrasting prediction qualities to examine whether it depends critically on a high-capacity predictor: a degraded variant MAPS-160M that uses the lightweight SP-160M, and an Oracle scheduler that uses ground-truth output lengths as a reference. We report E2E P99 and E2E mean latency on

LLaMA3.1-8B under the ShareGPT workload and Qwen2.5-3B under the BurstGPT workload.

**MAPS Remains Strong under Degraded Prediction.** Despite the substantially lower accuracy, MAPS-160M consistently outperforms all baselines, improving E2E P99 latency by 21.5% / 21.0% / 56.8% and mean latency by 12.9% / 6.6% / 59.1% over SGLang, vLLM, and Llumnix, respectively. The gap to MAPS is also modest, benefiting from the downstream UAC and scheduling design.

**MAPS Approaches Oracle.** MAPS closely tracks Oracle in tail latency, with E2E P99 within a relative gap of 19.8% that further narrows under high concurrency. Interestingly, MAPS achieves slightly lower mean latency than Oracle. This may be attributed to the oracle's exact allocation with perfect memory fitting, which can restrict long requests to a limited set of feasible decoders and potentially affect load balance. In contrast, imperfect prediction may allow long requests to be dispatched more broadly, albeit with the cost of increased contention, which affects tail latency. Overall, MAPS attains near-oracle performance, suggesting limited headroom for further predictor improvement.

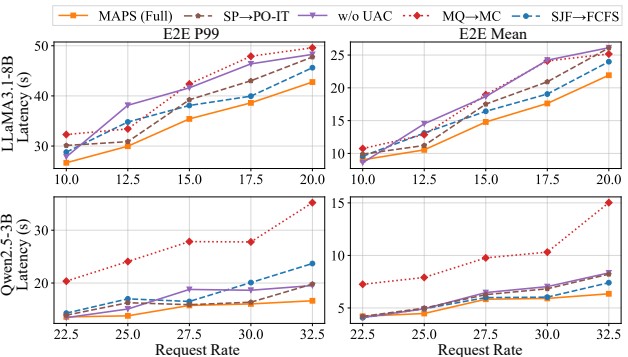

*Figure 6.* Ablation of MAPS on ShareGPT workloads.

### 4.4. Ablation Study

We conduct a comprehensive ablation study to quantify the contribution of each component in MAPS, as shown in Figure 6. All ablations are evaluated on real ShareGPT workloads under varying request rates, using E2E latency (mean and P99) to assess how each module contributes to latency reduction and efficient inference.

**SP→PO-IT.** We replace the SP with the PO-IT prediction baseline, while keeping all other components unchanged. Compared to MAPS, SP→PO-IT consistently incurs higher latency, increasing E2E P99 latency by 9.2% and E2E mean latency by 10.1% on average across both models and request rates. This demonstrates that prediction accuracy plays a critical role in guiding downstream memory-aware scheduling decisions. We further observe that under certain request rates, SP→PO-IT achieves performance close to MAPS,

suggesting that the proposed scheduling and calibration mechanisms are robust to different predictors. In particular, UAC helps maintain stable performance even when the predictor is less specialized.

**w/o UAC.** Removing UAC leads to a substantial increase in E2E latency, second only to MQ→MC among all ablations, with P99 and mean E2E latency increasing by 14.3% and 18.4%, respectively. This degradation arises because calibration improves system robustness by reducing underestimation of output length, which may otherwise lead to request preemption and cascading queue delays.

**MQ→MC.** We next examine the design of global routing by replacing the Minimum Queue (MQ) policy with a Maximum Capacity (MC) policy, which selects the decoder with the largest remaining KV cache capacity. Although MC appears intuitive from a memory-centric perspective, it significantly degrades both mean and tail latency under high load. On Qwen2.5-3B, P99 and mean E2E latency increase by up to $1.11\times$ and $1.36\times$, respectively. This result highlights that the primary objective of routing is to mitigate queue buildup rather than merely balance residual memory. MQ prioritizes decoders with shorter waiting queues while still enforcing memory feasibility, effectively limiting queue growth and yielding more stable latency performance.

**SJF→FCFS.** We replace local SJF reordering with an FCFS policy. Removing SJF consistently increases both mean and tail latency, with a more pronounced impact on E2E P99 under higher request rates. This indicates that local reordering remains beneficial even after global routing filters infeasible or highly skewed requests.

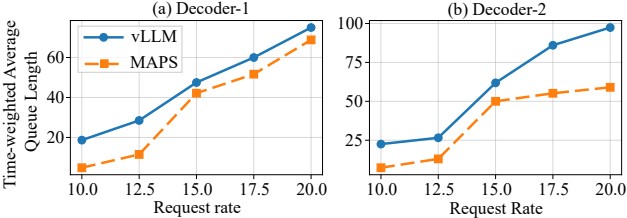

*Figure 7.* Time-weighted average queue length under different request rates for each decoder.

**Queueing Behavior and Waiting Reduction.** We use the widely adopted ShareGPT workload to evaluate the effectiveness of MAPS in reducing queueing. We compare MAPS against the competitive vLLM baseline and measure queueing pressure using the time-weighted average queue length from queueing theory over the entire execution, as shown in Figure 7. Across all request rates, MAPS consistently maintains shorter queues on both decoders. At moderate request rates (10 and 12.5 req/s), MAPS reduces the average queue length by 62.9% by dispatching requests to memory-feasible decoders with the shortest queues.

At higher loads (17.5 and 20 req/s), MAPS degrades more gracefully than vLLM and maintains near-balanced queueing across decoders: the average queue-length ratio between the two decoders is 0.96, close to perfect balance, compared to $1.36\times$ under vLLM. Importantly, queue reduction is observed on both decoders rather than being achieved by shifting load between instances, demonstrating that MAPS mitigates waiting through predictive, memory-aware scheduling instead of reactive load balancing.

## 5. Limitations

**Deployment scale.** Our evaluation is conducted on a cluster of six NVIDIA A6000 GPUs in a PD-disaggregated setup with one prefiller and two decoders, which is sufficient to expose the scheduling challenges MAPS targets. The architecture is designed to scale, relying on lightweight per-decoder metadata and asynchronous state collection that keep scheduling overhead at 1 ms even under high load. Empirical validation on substantially larger deployments remains future work.

**Chain-of-thought workloads.** MAPS targets general LLM serving where total generation length is the primary cost driver. In chain-of-thought reasoning, intermediate tokens are dynamically generated and highly variable, making reliable length prediction an open problem. Importantly, this challenge lies in the prediction module rather than the scheduling framework: the core design of memory-aware routing, SJF, and UAC remains applicable, and future CoT-aware predictors can be seamlessly integrated.

**Predictor adaptation.** The speculative predictor is fine-tuned on Alpaca and evaluated on conversational and bursty workloads. When deployed on workloads with substantially different distributions, such as code generation or agentic tool-use traces, predictor accuracy may degrade and require additional adaptation, which UAC partially mitigates through online calibration.

## 6. Conclusion

In this paper, we have presented MAPS, a memory-aware predictive scheduling framework for large language model serving. MAPS combines a device-assisted speculative predictor with uncertainty-aware calibration to derive coverage-guaranteed output length upper bounds at low overhead. Leveraging these calibrated bounds, MAPS performs hierarchical global-local scheduling that jointly accounts for decoder memory availability and queueing dynamics. Extensive evaluations across multiple models and workloads have shown that MAPS substantially reduces tail latency and improves scheduling stability under real-world bursty workloads.

## Acknowledgements

We thank the anonymous reviewers for their constructive comments and valuable suggestions. This work was supported by the National Natural Science Foundation of China (Youth) under Grant No. 62306208 and No. 62502341; the China Postdoctoral Science Foundation (Certificate No. 2025M771588); the Postdoctoral Fellowship Program (Grade B) of the China Postdoctoral Science Foundation under Grant No. GZB20250410; the National Key R&D Program of China under Grant No. 2025YFB4506600; and the Tianjin Xinchuang Haihe Lab under Grant No. 22HHX-CJC00002.

## Impact Statement

This paper presents work whose goal is to advance the field of Machine Learning. There are many potential societal consequences of our work, none of which we feel must be specifically highlighted here.

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

# A. Motivation: Why Round-Robin Routing Fails under Heterogeneous LLM Workloads

This section provides a controlled simulation study to quantify how output-length heterogeneity can induce *persistent* and *skewed* KV-cache pressure across decoders under round-robin (RR) routing, even when requests are evenly distributed by count. The goal is not to perfectly reproduce a specific serving engine, but to isolate a minimal mechanism that explains why *request-count fairness* does not imply *memory-pressure fairness*, thereby motivating our memory-aware predictive scheduling framework.

We consider a PD-disaggregated serving deployment with one prefiller and $N$ decoders. Requests arrive to the system according to a Poisson process with rate $\lambda$ (req/s). Each request $r$ has an input length $L_p(r)$ and an output length $L_{\text{out}}(r)$ in tokens. We denote their empirical distributions by $P_p$ and $P_{\text{out}}$, estimated from a real workload trace (Wang et al., 2025b).

Each decoder $d \in \{1, \dots, N\}$ maintains a KV-cache pool with capacity $C$ tokens. Following the KV accounting commonly used in memory planning, we approximate the KV token footprint of a request $r$ on a decoder as

$$M(r) \triangleq L_p(r) + L_{\text{out}}(r). \tag{8}$$

This abstraction captures the dominant dependence of KV footprint on the total number of tokens that must be retained for attention during decoding. If desired, the token-based capacity $C$ can be translated to bytes via model-dependent constants (layers, heads, dtype). Our study uses the token capacity for clarity.

**Routing Policy.** Under RR routing, incoming requests are assigned to decoders cyclically:

$$d(r_i) = 1 + (i \bmod N), \tag{9}$$

where $i$ is the arrival index.

**Queueing-theoretic Perspective.** From a queueing perspective, RR routing with Poisson arrivals can be viewed as uniformly thinning the global arrival process into $N$ per-decoder streams, each with rate approximately $\lambda/N$. Each decoder thus behaves as an $M/G/1$ queue with processor sharing (PS), where the service requirement of a request is proportional to its output length. Specifically, the service time is given by $S = L_{\text{out}} \cdot t_{\text{tok}}$, where $t_{\text{tok}}$ denotes the average per-token decoding time. Since the output length $L_{\text{out}}$ follows a highly heterogeneous empirical distribution estimated from real workloads, the induced service-time distribution is general (non-exponential), corresponding to the $G$ in the $M/G/1$ model. Processor sharing approximates the time-sliced and batched decoding behavior in

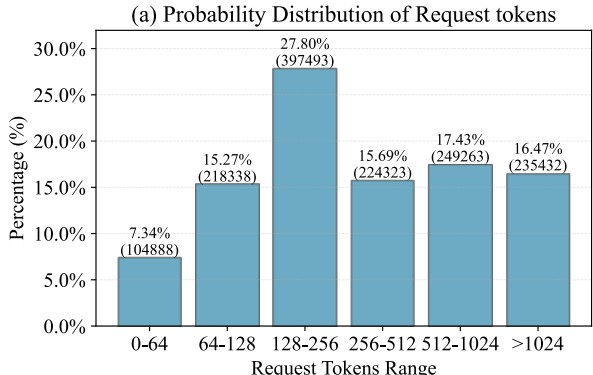

(a) Probability Distribution of Request tokens

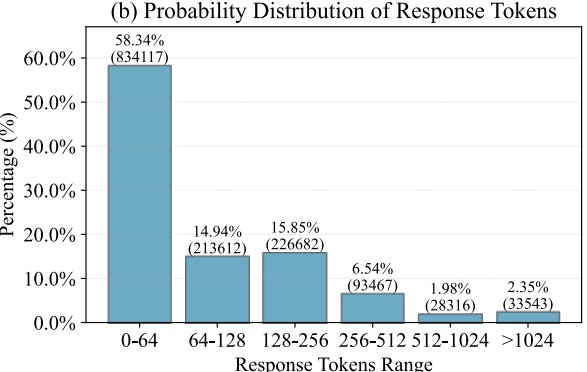

(b) Probability Distribution of Response Tokens

*Figure 8.* Empirical token-length distributions derived from the BurstGPT trace.

modern LLM engines. Under such general, heavy-tailed service-time distributions, classical queueing theory predicts that busy periods and congestion episodes can become long and highly variable, even when the average load is evenly balanced across decoders. Our simulation instantiates this minimal $M/G/1$-PS abstraction and further incorporates a finite KV-cache capacity constraint to study sustained and skewed memory-pressure dynamics. Thus, the M/G/1-PS view is used to explain the compute-side service variability, while our simulation extends it with a finite-capacity KV admission gate to capture memory-side congestion.

**Request Queueing.** Let $K_d(t)$ denote the number of KV tokens currently occupied on decoder $d$ at time $t$, and let $M(r)$ represent the KV cache demand of request $r$. A request $r$ can be admitted to decoder $d$ at time $t$ if

$$K_d(t) + M(r) \leq C, \tag{10}$$

where $C$ is the KV capacity of the decoder. Otherwise, the request is enqueued into a First-In-First-Out (FIFO) waiting queue at the decoder and admitted only when sufficient capacity becomes available.

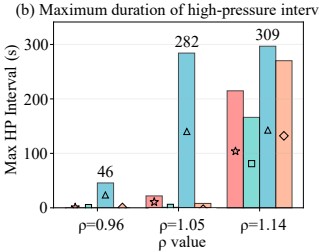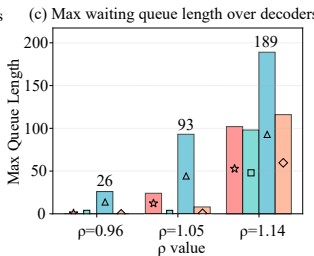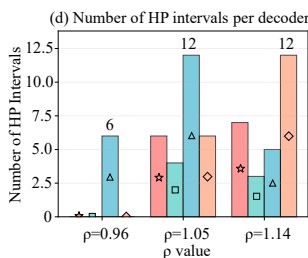

**Figure 9.** High-pressure dynamics under RR routing across utilization $\rho$, revealing persistent and uneven KV cache pressure across decoders, where some decoders experience prolonged high-pressure states while others remain lightly loaded.

## A.1. High-Pressure Interval Metric

Classic "busy period" in queueing theory is defined by whether the server is non-empty. For LLM decoding, however, the practical instability is often triggered when KV cache usage approaches the capacity limit, leading to waiting and/or preemption. We therefore define a memory-centric congestion metric.

**Definition A.1** (High-Pressure Interval, HP Interval). Let the KV cache utilization of decoder $d$ at time $t$ be

$$u_d(t) \triangleq \frac{K_d(t)}{C}. \tag{11}$$

Given a utilization threshold $\theta \in (0, 1)$, decoder $d$ is in the *high-pressure state* at time $t$ if $u_d(t) \geq \theta$. A *high-pressure interval* is a maximal contiguous time interval $[t_s, t_e)$ such that $u_d(t) \geq \theta$ for all $t \in [t_s, t_e)$.

For each decoder $d$, we characterize its high-pressure dynamics by measuring the total time spent in the high-pressure state $T_{\mathrm{hp}}^{(d)}$, the maximum duration of high-pressure intervals $B_{\mathrm{hp}}^{(d)}$, the number of high-pressure intervals $C_{\mathrm{hp}}^{(d)}$, and the corresponding time fraction $f_{\mathrm{hp}}^{(d)} = T_{\mathrm{hp}}^{(d)}/T_{\mathrm{sim}}$, where $T_{\mathrm{sim}}$ denotes the simulation horizon from the first request arrival to the last completion.

## A.2. Simulation Setup

We simulate 10,000 requests with:

- **Decoders:** $N = 4$.
- **KV cache capacity:** $C = 50{,}000$ tokens per decoder.
- **Token inference time:** $t_{\mathrm{tok}} = 1\,\mathrm{ms/token}$.
- **High-pressure threshold:** $\theta = 0.85$.
- **Workload:** The prompt length $L_p$ and output length $L_{\mathrm{out}}$ are sampled i.i.d. from the empirical length distributions measured from the BurstGPT trace. Specifically, we discretize both $L_p$ and $L_{\mathrm{out}}$ into multiple length ranges and sample them according to the corresponding probabilities shown in Figure 8.

## A.3. Results and Interpretation

To characterize when RR routing becomes vulnerable to persistent memory pressure, we use the standard queueing utilization $\rho \approx \lambda \mathbb{E}[L_{\mathrm{out}}] t_{\mathrm{tok}}/N$. While $\rho$ reflects compute-side saturation in decoding, our high-pressure intervals quantify memory-side congestion under finite KV capacity. In our evaluation, we vary the arrival rate $\lambda$ to span a range of utilization regimes and analyze system behavior under different queueing load conditions.

Figure 9 provides a controlled validation of our queueing-theoretic analysis, demonstrating that RR routing can induce *persistent and highly skewed memory pressure* across decoders, even when requests are evenly distributed by count. Although RR ensures identical arrival rates in expectation, the heavy-tailed output-length distribution causes substantial divergence in decoder-level KV-cache occupancy over time.

In our simulation, we fix a single realization of the workload (arrival sequence and input/output lengths) and vary only the load intensity, so as to isolate the impact of utilization on system dynamics. Under this fixed workload, RR routing deterministically assigns each decoder a specific subsequence of requests, causing the same decoder (decoder-3, $d_3$, in our setup) to consistently emerge as the most heavily loaded one across different utilization levels. This behavior reflects a structural imbalance amplification effect of RR routing under heterogeneous generation lengths, rather than randomness or decoder-specific properties. As shown in Figure 9(a), even at a moderate load ($\rho \approx 0.96$), $d_3$ already enters the high-pressure state for a non-negligible fraction of time, while other decoders remain largely uncongested. When the system operates near critical utilization ($\rho \approx 1.05$), $d_3$ spends nearly 80% of the execution time above the high-pressure threshold, whereas the remaining decoders experience only sporadic or short-lived pressure.

This imbalance becomes more pronounced as $\rho$ slightly exceeds one. Figure 9(b) and 9(c) show that $d_3$ not only suffers the longest continuous high-pressure intervals, exceeding 280 seconds, but also accumulates the largest waiting queues, despite identical routing and capacity. In

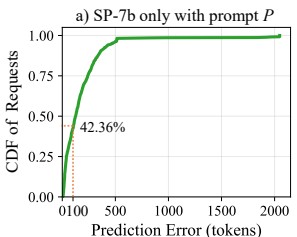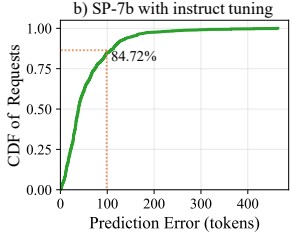

*Figure 10.* Cumulative distribution function (CDF) of output-length prediction error for SP-7B under different adaptation settings. Prediction error is measured as the absolute difference between predicted and ground-truth output length, in tokens.

contrast, other decoders exhibit significantly shorter high-pressure durations and smaller queue buildup. Figure 9(d) further confirms that $d_3$ experiences more frequent high-pressure episodes, indicating repeated and sustained congestion rather than isolated spikes.

These results empirically confirm a key implication of the $M/G/1$-PS abstraction: under heavy-tailed service requirements, utilization parity does not imply memory-pressure fairness. RR routing equalizes request counts but ignores variability in KV cache demand, allowing a single decoder to become persistently overloaded while others remain underutilized. This structural imbalance can persist even when $\rho < 1$ and worsens rapidly as the system approaches saturation, motivating the need for memory-aware and prediction-guided scheduling.

## B. Training Details of the Speculative Predictor

The speculative predictor in MAPS is instantiated at two scales to accommodate the heterogeneous resource budgets of edge devices. SP-7B uses Vicuna-7B, initialized from the LLaMA-2-7B checkpoint, while SP-160M uses the lightweight Vicuna-160M, which targets devices with more constrained memory. Both variants are adapted via parameter-efficient LoRA fine-tuning. The training data consist of 20K prompts randomly sampled from the Alpaca dataset, where the first 10K prompts are used for fine-tuning and the remaining 10K prompts are reserved for evaluation. The same data split is applied to both variants to ensure a fair comparison.

For each training prompt, the supervision target is constructed by querying multiple downstream LLMs, including LLaMA3.1-8B and Qwen2.5-3B, under four decoding temperatures (0.0, 0.3, 0.5, and 0.7). The maximum output length observed across all model-temperature combinations is used as the ground-truth target. This conservative formulation is designed to upper-bound generation variability across heterogeneous models and decoding behaviors, ensuring safety when predictions are used for memory-aware

scheduling.

We apply LoRA adaptation with rank 16 and scaling factor 32, injecting LoRA modules into all linear projection layers, including attention and feed-forward components. Training is performed for 3 epochs using the AdamW optimizer with a learning rate of $2 \times 10^{-5}$ and a cosine learning-rate schedule with a warm-up ratio of 0.03. The effective batch size is 64, achieved via a per-device batch size of 2 and gradient accumulation over 16 steps. Training uses bfloat16 precision with TF32 enabled for numerical stability.

During inference, the predictor employs greedy decoding with a tightly constrained output length, incurring negligible overhead and allowing prediction to be overlapped with cloud-side prefill execution. We analyze the distribution of output-length prediction errors produced by SP-7B under different adaptation settings. Figure 10 reports the cumulative distribution functions (CDFs) of prediction error for (i) a prompt-only predictor using an off-the-shelf Vicuna-7B model with the prediction prompt $P$, and (ii) the same model after LoRA-based parameter-efficient adaptation.

Without parameter adaptation, the prompt-only predictor exhibits a heavy-tailed error distribution. Only 42.36% of requests achieve an absolute prediction error within 100 tokens (Acc-100), and prediction errors frequently reach the predefined truncation limit of 2048 tokens. After LoRA-based adaptation, the error distribution becomes substantially more concentrated: Acc-100 increases to 84.72%, and the maximum prediction error is reduced to within 500 tokens.

This result indicates that lightweight parameter-efficient adaptation is crucial for mitigating extreme tail errors that cannot be addressed by prompt design alone. By significantly stabilizing the error distribution, LoRA adaptation provides a reliable foundation for subsequent uncertainty-aware calibration and memory-aware scheduling.

## C. Global Memory-Aware Scheduling Algorithm

Algorithm 1 provides an implementation-level description of MAPS's global memory-aware scheduling logic. While the core scheduling decisions are discussed in Section 3.3, we highlight several additional design considerations here for completeness.

First, although the global admission decision uses a single block budget $B_i^{\text{need}}$ that covers both prompt and predicted output tokens, MAPS internally tracks the prompt-related and output-related components separately during the request lifecycle. Once prefilling completes and the prompt KV cache is transferred to the selected decoder, the prompt-related portion of the reservation is released. The remaining

## Algorithm 1 Global Memory-Aware Scheduling

**Input:** Job $J_i$, prompt length $L_i^{\text{prom}}$, block size $S$, timeout $\beta$, runtime state collector (RSC)

1: Query RSC to obtain decoder states $\mathcal{D}$, including $\{(B_k^{\text{eff}}, Q_k)\}_{d_k \in \mathcal{D}}$
2: Try to fetch the calibrated length upper bound $\hat{L}_i^{\text{up}}$ from RSC with a timeout of $\beta$
3: **if** $\hat{L}_i^{\text{up}}$ is unavailable within $\beta$ **then**
4:     **return RoundRobin**$(J_i)$
5: **end if**
6: $B_i^{\text{need}} \leftarrow \left\lceil (L_i^{\text{prom}} + \hat{L}_i^{\text{up}})/S \right\rceil$
7: $\mathcal{D}_{\text{feasible}} \leftarrow \emptyset$
8: **for** each decoder $d_k \in \mathcal{D}$ **do**
9:     **if** $B_i^{\text{need}} \leq B_k^{\text{eff}}$ **then**
10:         $\mathcal{D}_{\text{feasible}} \leftarrow \mathcal{D}_{\text{feasible}} \cup \{d_k\}$
11:     **end if**
12: **end for**
13: **if** $\mathcal{D}_{\text{feasible}} \neq \emptyset$ **then**
14:     $d_\star \leftarrow \arg\min_{d_k \in \mathcal{D}_{\text{feasible}}} Q_k$
15: **else**
16:     $d_\star \leftarrow \arg\max_{d_k \in \mathcal{D}} B_k^{\text{eff}}$
17: **end if**
18: **Dispatch**$(d_\star, J_i)$

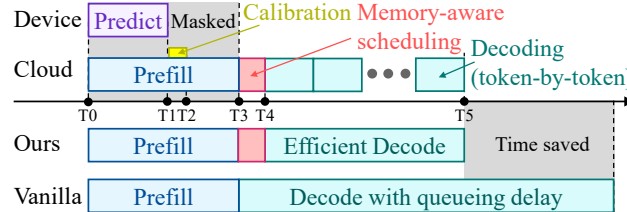

*Figure 11.* Timeline analysis of MAPS.

portion, corresponding to the predicted output, is held until generation finishes, at which point the reserved blocks are reclaimed. This staged reservation and release mechanism avoids long-lived over-reservation and improves effective memory utilization under high concurrency.

Second, the DISPATCH operation abstracts the actual request handoff to the selected decoder $d_\star$. In practice, this step resolves the network endpoint (e.g., IP address and service port) of $d_\star$ and forwards the request accordingly. The scheduling logic itself remains agnostic to transport details, allowing MAPS to be integrated with different serving backends and communication protocols.

Together, these implementation details ensure that the scheduling policy described in Section 3.3 can be realized efficiently and safely in a practical PD-disaggregated LLM serving system.

## D. Timeline Analysis

To provide a temporal understanding of the latency benefits of MAPS, we analyze the end-to-end inference workflow from a timeline perspective. As illustrated in Figure 11, the inference lifecycle is divided into three stages: overlapping prefill and prediction, memory-aware scheduling, and decoding with SJF reorder.

**Overlapping Prefill and Prediction.** Upon request submission at time $T_0$, the input prompt is transmitted to the cloud

to initiate the prefill phase, while the request-origin device concurrently performs speculative prediction.

Let $t_{\text{req}}$ denote the prompt transmission latency, $t_{\text{prefill}}$ the cloud-side prefill time (T0–T3), $t_{\text{pred}}$ the device-side prediction latency (T0–T1), and $t_{\text{cali}}$ the calibration latency (T1–T2). The total latency of this stage is

$$T_{\text{pre}} = \max(t_{\text{req}} + t_{\text{prefill}},\ t_{\text{pred}} + t_{\text{cali}}). \quad (12)$$

In practice, speculative prediction incurs negligible overhead due to the lightweight model and the absence of data transfer, while calibration involves only lightweight postprocessing with $O(n \log n)$ complexity. As a result, prediction and calibration are fully masked by prefilling in most cases, i.e., $t_{\text{pred}} + t_{\text{cali}} \ll t_{\text{req}} + t_{\text{prefill}}$, consistent with the measurements in Section D.1.

**Memory-Aware Scheduling.** After prefill completes, MAPS performs memory-aware scheduling based on the calibrated upper bound of the predicted output length. This step introduces a scheduling overhead $t_{\text{sched}}$ (T3–T4), which involves only lightweight arithmetic and state lookups and is negligible in practice, consistent with the measurements in Section D.1.

**Decoding Phase.** Let $t_{\text{gen}}$ denote the hardware-limited time to generate the output tokens once decoding proceeds without stalls. In vanilla systems, request-agnostic routing policies (e.g., RR) may dispatch requests to already congested or memory-pressured decoders, introducing additional delay from (i) queueing before admission and (ii) decoding stalls such as preemption or KV-cache swapping. We aggregate these effects as $t_{\text{delay}}$, yielding

$$T_{\text{decode}}^{\text{vanilla}} = t_{\text{gen}} + t_{\text{delay}}. \quad (13)$$

By contrast, MAPS prioritizes memory feasibility at dispatch time using calibrated bounds and routes requests to lightly queued decoders, largely avoiding severe delays during decoding. Consequently, the decoding latency approaches the hardware limit:

$$T_{\text{decode}}^{\text{MAPS}} \approx t_{\text{gen}}. \quad (14)$$

**End-to-End Latency Comparison.** The end-to-end latency

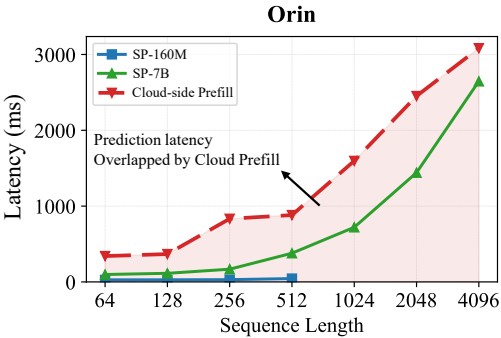

*Figure 12.* Runtime latency comparison between device-side speculative prediction and cloud-side prefill.

of a request can be decomposed as:

$$T_{\text{E2E}}^{\text{vanilla}} = t_{\text{req}} + t_{\text{prefill}} + t_{\text{gen}} + t_{\text{delay}}, \tag{15}$$

$$T_{\text{E2E}}^{\text{MAPS}} = t_{\text{req}} + t_{\text{prefill}} + t_{\text{sched}} + t_{\text{gen}}. \tag{16}$$

Accordingly, the net latency reduction is

$$\Delta T_{\text{E2E}} = T_{\text{E2E}}^{\text{base}} - T_{\text{E2E}}^{\text{MAPS}} = t_{\text{wait}} - t_{\text{sched}}. \tag{17}$$

Under moderate to high load, queueing and preemption overheads dominate decoding latency ($t_{\text{wait}} \gg t_{\text{sched}}$), enabling MAPS to achieve a net reduction in end-to-end latency by mitigating queueing delays and preemption during decoding.

### D.1. Negligible Runtime Overhead

In this section, we measure the runtime overhead introduced by MAPS and show that the additional prediction and scheduling incur near-zero overhead relative to the original LLM inference pipeline.

**Runtime Overhead of Device-side Prediction.** We evaluate the runtime overhead introduced by device-side speculative prediction on NVIDIA Jetson Orin NX. We sweep the input sequence length from 64 to 4096 tokens and compare the device-side prediction latency with the cloud-side prefill latency at each length. All reported results are obtained by averaging eight independent runs to ensure stability and reproducibility.

As shown in Figure 12, across both predictor variants, device-side prediction is fully overlapped by the cloud-side prefill stage. The lightweight SP-160M incurs only tens of milliseconds of prediction latency across the entire range of input lengths, while the larger SP-7B exhibits higher latency, reaching several hundred milliseconds for long inputs on Orin NX. In contrast, the cloud-side prefill latency consistently operates at a substantially larger time scale, ranging from hundreds of milliseconds to several seconds as sequence length increases. As a result, speculative prediction does not appear on the critical path and incurs no measurable increase in TTFT or E2E latency.

In the worst event that prediction exceeds prefilling latency, MAPS safely defers scheduling until the calibrated bound becomes available. If the waiting time exceeds a predefined threshold, the scheduler falls back to RR routing, ensuring bounded delay and preserving system liveness. This fallback mechanism guarantees that MAPS remains robust even when prediction overlap cannot be fully achieved.

**Runtime Overhead of Scheduling.** MAPS employs a hierarchical scheduling design that consists of a global memory-aware scheduling (MAS) and a local SJF reorder within each decode instance. The local Reorder module only performs lightweight queue reordering based on calibrated upper bound and does not involve complex computation. As a result, its overhead is negligible and remains well below the millisecond level in practice. We therefore focus our analysis on the runtime overhead of the MAS, which is responsible for instance-level dispatch decisions.

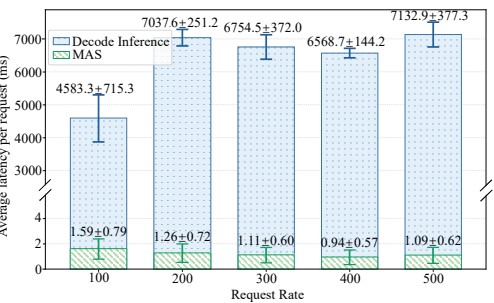

*Figure 13.* Runtime latency comparison between memory-aware scheduling and decode inference.

Figure 13 reports the per-request latency of MAS and decode inference under varying incoming request rates, measured over 1,000 requests for each setting. Across all evaluated loads, MAS consistently incurs only 1–2 ms of latency per request, with limited variance as the request rate increases.

In contrast, decode inference dominates the end-to-end runtime. The broken y-axis visualization further highlights this disparity, emphasizing that dispatch latency is negligible relative to decoding. Together, these results demonstrate that the hierarchical scheduling design in MAPS introduces negligible runtime overhead in practice, while enabling effective memory-aware routing decisions at scale.

### E. Parameter Sensitivity Analysis of Maximum Waiting Time

We analyze the sensitivity of MAPS to the maximum waiting time parameter, which prevents long-running requests from starving due to short-job prioritization. Figure 14 reports both E2E mean latency and P99 latency under two representative request rates (10 and 20 req/s), as the max

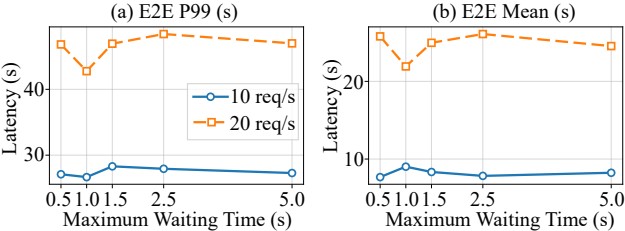

*Figure 14.* Sensitivity of MAPS to the maximum waiting time.

wait time varies from 0.5s to 5.0s.

Overall, MAPS exhibits stable latency performance across a wide range of maximum waiting time values. For both request rates, neither the mean latency nor the tail latency shows monotonic degradation as the wait time increases. Instead, latency remains within a narrow band, indicating that MAPS is largely insensitive to this parameter once it lies within a reasonable operating regime.

This robustness stems from the predictive, memory-aware design of MAPS. Since requests are dispatched only to memory-feasible decoders based on calibrated length estimates, decoder queues are protected from early saturation by long-generation jobs. As a result, batching efficiency and queueing behavior are no longer dominated by the maximum waiting time threshold. Consequently, maximum waiting time acts as a secondary tuning knob rather than a critical determinant of system performance. Based on these observations, we fix the maximum waiting time to 1.0s in our main experiments, as it lies within the stable region and provides a good balance between batching opportunity and responsiveness.

