# OpenReview forum: "MAPS: Memory-Aware Predictive Scheduling Framework for Large Language Model Serving"
_ICML.cc/2026/Conference — ICML 2026 regular_

### Official Review · Reviewer_NrUJ · 2026-02-19

**Soundness:** 2
**Presentation:** 3
**Significance:** 2
**Originality:** 2
**Overall Recommendation:** 2
**Confidence:** 3

**Summary:**

During PD disaggregation, decode instances suffer from load imbalance because the output lengths of different requests are random and unpredictable.
MAPS introduces a predictor module on the client side that forecasts output lengths based on user inputs.
The prediction assists request scheduling on the cloud side.
Furthermore, MAPS incorporates an uncertainty-aware calibration module to refine the prediction results.
Experimental results demonstrate that MAPS effectively reduces latency and improves throughput through precise scheduling.

**Compliance With Llm Reviewing Policy:**

Affirmed.

**Final Justification:**

I have read the authors' rebuttal and the subsequent discussion. My concerns regarding prediction error (W2) and robustness under inaccurate predictions (W3) have been resolved by the additional experiments. However, my primary concern about client-side prediction (W1) remains. In the follow-up discussion, the authors acknowledged that cloud-side prediction is a feasible design and that their framework is "fundamentally agnostic to the deployment location of prediction." This effectively concedes that client-side prediction — listed as the first core contribution of the paper — is not a necessary component of the system.

I maintain my score of 2 (Reject). Once client-side prediction is removed as a contribution, the remaining technical content (i.e., early prediction signals for memory-aware scheduling and SJF-style prioritization) represents a reasonable but incremental improvement over existing PD disaggregation systems, which I believe falls below the novelty threshold for ICML. The paper would benefit from either presenting a more compelling case for why prediction placement genuinely matters as a systems contribution, or refocusing on the scheduling and calibration components with stronger differentiation from prior work.

**Key Questions For Authors:**

1. Why is the prediction performed on the client side instead of the cloud side concurrently with the prefill stage of the request?
2. Have the authors attempted to validate the effectiveness of the predictor on more diverse datasets?

**Limitations:**

yes

**Strengths And Weaknesses:**

Strengths
1. The prediction of output length significantly improves load balancing, while the uncertainty-aware calibration provides a safety upper bound.
2. The algorithmic design in the paper is sound and reasonable.

Weaknesses
1. The idea of performing prediction on the client side seems unnecessary and increases user-side overhead. I think it would be better to perform the prediction on the cloud. It can be done concurrently with the prefill stage of the request since the result is only needed for decode.
2. The error of the predictor seems large. The average output length of the Alpaca dataset may only be around 300 tokens.
In addition, there is a lack of in-depth analysis regarding the predictor's performance in predicting output lengths on other datasets.
3. It is expected to see system performance experiments conducted when predictions are inaccurate.

---

> ### Author Rebuttal · Authors · 2026-03-31
>
> We thank the reviewer for the constructive feedback and for recognizing the soundness and reasonableness of our design. Below we address the main concerns.
> ```
> W1&Q1: Why not perform prediction on the cloud?
> ```
> Client-side prediction is essential for true latency hiding and avoiding resource contention.
>
> (1) **Cloud-side prediction is not free to overlap.** Prefill is already compute-intensive and often saturates GPU resources, so adding a prediction model introduces additional contention. Moreover, cloud-side prediction aggregates requests from all clients; under high concurrency, prediction tasks are queued with other workloads. In our measurements on ShareGPT, cloud-side prediction latency reaches **15.99 s on average**, which cannot be effectively hidden by prefill.
>
> Even with additional GPUs, avoiding such queueing would require scaling prediction resources proportionally with request volume, which introduces significant deployment overhead and cost.
>
> (2) **Advantages of client-side prediction.** Prediction starts immediately without network delay or queueing, and runs independently of cloud workload. Moreover, it naturally scales with the number of requests by leveraging existing device resources, without requiring additional cloud provisioning.
>
> (3) **Aligned with emerging edge–cloud trends.** Modern client devices are increasingly capable (e.g., GPU-equipped PCs and smartphones with AI accelerators), and both academia and industry are actively exploring offloading lightweight components to the edge. Similar to PD disaggregation, our design decouples prediction from inference across client and cloud, enabling better resource utilization and reduced contention under high concurrency.
>
> Overall, cloud-side prediction introduces contention and delay, while client-side prediction enables reliable latency hiding and timely scheduling signals.
> ```
> W2&Q2: Prediction accuracy and generalization.
> ```
> We clarify that the prediction error is moderate and expected under inherent uncertainty.
>
> On Alpaca, the predictor achieves an MAE of 56 tokens. Given an average output length of ~300 tokens, this corresponds to a relative error of ~**18.7%**, which is reasonable for this task and sufficient for guiding scheduling decisions. Output length prediction is inherently uncertain: even for the same prompt, different temperatures or runs can produce substantially different lengths.
>
> To reflect this uncertainty, our supervision target is not a single reference output. Instead, we construct a conservative target by querying multiple models under different decoding temperatures and taking the maximum length (Appendix B), which increases both the scale and variance of the target. As a result, some prediction error is unavoidable.
>
> More importantly, MAPS does not rely on exact prediction accuracy. Scheduling is primarily sensitive to underestimation, which can lead to memory overcommitment. MAPS addresses this via uncertainty-aware calibration, which continuously adjusts prediction errors online and ensures robust scheduling.
>
> To evaluate generalization, we further test the predictor on diverse datasets. As shown below, performance remains stable across distributions. Detailed error distributions see: https://anonymous.4open.science/r/Maps-09B3/NrUJ/Q2.md.
> |Dataset|MAE↓|Acc-50↑|Acc-100↑|
> |-|-|-|-|
> |ShareGPT|70.7|46.1%|75.7%|
> |LMSYS-Chat-1M|89.3|74.3%|85.6%|
> |UltraChat|61.7|88.5%|89.4%|
> ```
> W3: Robustness to inaccurate predictions
> ```
> We evaluate MAPS under varying prediction quality and show that it remains robust even with degraded predictors.
>
> We consider three levels: (1) SP-160M (a degraded predictor using a much smaller Vicuna-160M model), (2) SP (default), and (3) Oracle (using ground-truth output length as an upper bound).
>
> Despite substantially lower prediction accuracy (e.g., MAE increases by **62.8%**), MAPS with SP-160M still achieves strong system performance: E2E P99 latency degrades moderately (**13.2%**), while consistently outperforming **all** baselines (e.g., vLLM) across workloads. This demonstrates that MAPS remains effective even under significantly degraded prediction quality. Detailed results are available at: https://anonymous.4open.science/r/Maps-09B3/NrUJ/W3.md.
>
> This indicates that MAPS does not rely on highly accurate prediction. UAC mitigates underestimation risk, and SJF primarily depends on relative ordering rather than exact values.
>
> For completeness, we also evaluate the Oracle setting. While Oracle achieves the lowest P99 latency, the gap remains small, and MAPS achieves comparable or slightly better mean latency.
>
> Overall, MAPS maintains strong performance across a wide range of prediction quality, demonstrating robustness to inaccurate predictions.
>
> ---
> We hope that our clarifications help resolve your concerns and demonstrate the strengths of our approach. We would sincerely appreciate your consideration in raising the score. Please feel free to ask if anything is unclear.

---

> > ### Author Rebuttal · Reviewer_NrUJ · 2026-04-04
> >
> > I appreciate the authors' detailed response and the additional experiments. My concerns regarding W2 and W3 have been resolved. However, I still have reservations about W1. In my view, client-side prediction is impractical for the following reasons:
> > 1. The fundamental value of cloud GPUs lies in consolidating computation from many clients to improve resource utilization. Compared to running a prediction model independently on each user device, batching predictions on the cloud is clearly more resource-efficient. Regarding the concern that prediction demand scales with request volume — the same is true for inference itself, which consumes far more GPU resources. Prediction overhead is minor relative to actual serving, and the two tasks scale at the same rate.
> > 2. It is unrealistic to assume that users are willing to dedicate scarce on-device compute to running a prediction model. Model loading and inference increase power consumption and interfere with other tasks, solely to benefit cloud-side scheduling — offering no direct value to the user.
> > 3. I acknowledge that edge-cloud collaboration is a meaningful direction. However, I find the case for client-cloud collaboration less convincing. If the paper were repositioned to leverage edge servers (i.e., edge-side infrastructure) rather than end-user devices for prediction, the design would be more practical and compelling.
> >
> > Since client-side prediction is a core contribution of this paper, I still tend to maintain my current score.

---

> > > ### Author Response · Authors · 2026-04-04
> > >
> > > Thank you for the thoughtful discussion. We agree that cloud-side prediction is a feasible design. With sufficient dedicated resources and coordinated batching, a lightweight predictor can be efficiently executed on the cloud and may overlap with prefill.
> > >
> > > Importantly, our framework is fundamentally **agnostic to the deployment location of prediction**. The speculative predictor (SP) defines how to predict, while RSC and hierarchical scheduling depend only on the **timeliness and validity of the prediction result**, not where it is computed. As a result, cloud-side, edge-side, and device-side deployments are all compatible with our design.
> > >
> > > We also view **on-device collaboration** as a promising direction and actively explore this setting. Such designs are increasingly adopted in both industry and research. For example, **Apple and Qualcomm have integrated on-device AI capabilities** (e.g., Apple Neural Engine and Snapdragon NPUs), enabling efficient local inference. Therefore, cloud-side and device-side prediction are both feasible solutions with different system trade-offs.
> > >
> > > Regarding the concern that device-side prediction offers no direct value to users, we clarify that our objective is **user-facing performance**, namely reducing end-to-end latency and improving responsiveness. Prediction improves scheduling decisions, which directly reduces queueing delay and leads to faster responses. Therefore, it provides **direct value to end users**, rather than serving only system-side optimization.
> > >
> > > From a system perspective, the key issue is not where prediction is placed, but whether it can provide timely prediction signals without interfering with the serving path, and who bears the associated resource cost.
> > >
> > > - **Cloud-side prediction**. It requires additional infrastructure beyond the serving path. Its latency depends on aggregated workloads and system dynamics, and thus timely availability is not inherent but depends on provisioning and scheduling assumptions.
> > >
> > > - **Edge-side prediction**. This similarly requires additional deployment and resource provisioning. Requests are still aggregated, and prediction latency is influenced by shared-resource queueing, although network distance may be reduced.
> > >
> > > - **Device-side prediction**. Prediction starts immediately when the request is formed and runs in parallel with request transmission and cloud prefill. Each request is processed independently, without multi-tenant aggregation or queueing. This provides a natural head start and a **robust** way to ensure that prediction is available before scheduling, without introducing dependencies on shared infrastructure. The required compute is lightweight and tied to a single request. In addition, this design **avoids** an extra network dependency for prediction, further simplifying the system.
> > >
> > > We also agree that edge-side deployment is a meaningful alternative. Compared to device-side prediction, it still introduces an additional transmission and aggregation stage. As on-device capabilities (e.g., NPUs) continue to improve, the boundary between edge and end devices is becoming increasingly **blurred**.
> > >
> > > Finally, we emphasize that our main contribution lies in enabling **early and reliable prediction signals to support memory-aware scheduling and SJF-style prioritization**, thereby reducing end-to-end latency. **Device-side prediction** is one natural instantiation that achieves this goal with minimal additional system complexity.

---

### Official Review · Reviewer_PTKc · 2026-02-22

**Soundness:** 3
**Presentation:** 2
**Significance:** 2
**Originality:** 3
**Overall Recommendation:** 4
**Confidence:** 3

**Summary:**

This paper proposes MAPS, a memory-aware predictive scheduling framework for prefill–decode (PD) disaggregated LLM serving. MAPS overlaps device-side speculative output-length prediction with cloud-side prefill, calibrates predictions with a one-sided conformal predictor to produce coverage-guaranteed upper bounds, and performs hierarchical scheduling: inter-instance routing by memory feasibility and minimum queue length, and intra-instance SJF reordering to reduce head-of-line blocking. Experiments on two models and two workloads report substantial mean and tail latency reductions over vLLM, SGLang, and Llumnix, with ablations attributing gains to prediction, calibration, and hierarchical scheduling.

**Compliance With Llm Reviewing Policy:**

Affirmed.

**Final Justification:**

After reviewing the feedback and discussions, I maintain my score.

**Key Questions For Authors:**

1. The predictor is trained/evaluated on Alpaca, but scheduling results use ShareGPT/BurstGPT. How well does the predictor generalize across these distributions in practice?
2. The global router selects the feasible decoder with the minimum queue length. Did you consider objectives that also account for batching efficiency or the earliest available start time? How sensitive are results to this choice under high concurrency?
3. Does the speculative prediction still finish before the cloud-side prefill when the input prompt is extremely short but requires a long, complex output?
4. What is the specific network bandwidth impact of the Runtime State Collector periodically aggregating metadata from many decoders in a large-scale cluster?

**Limitations:**

yes

**Strengths And Weaknesses:**

Strengths:
1. The paper introduces a device-assisted speculative predictor to hide prediction latency by overlapping with prefill; this is a pragmatic and underexplored angle for length-aware routing in PD settings.
2. The paper provides meaningful results and includes informative ablations.
3. The system architecture and scheduling logic are clearly described
Weaknesses:
1. Artificial heterogeneity via memory caps could bias results.
2. Core scheduling evaluation is on one prefiller and two decoders (six GPUs total), which lacks multi-node or larger-cluster experiments and leaves scalability and robustness unclear.
3. There are some clarity or presentation issues.

---

> ### Author Rebuttal · Authors · 2026-03-31
>
> We thank the reviewer for the insightful feedback and for recognizing our design and empirical results. We address the concerns below with additional clarification.
> ```
> W1: On Heterogeneity
> ```
> We clarify that memory caps are not introduced to bias results, but to create a controlled setting for studying memory heterogeneity.
>
> In practice, heterogeneous memory availability naturally arises in LLM serving due to long-running requests, varying output lengths, and KV cache accumulation, leading to fragmentation and load divergence across decoders. This behavior is empirically observed (Fig. 1(b)) and theoretically supported (Appendix A). However, such heterogeneity is dynamic and difficult to reproduce consistently.
>
> To enable controlled and reproducible evaluation, we introduce fixed memory caps to emulate heterogeneous resource conditions. This setting reflects realistic system behavior while amplifying heterogeneity, allowing us to systematically evaluate scheduling performance under imbalance, rather than introducing artificial bias.
> ```
> W2: On Scalability
> ```
> Please see Reviewer 2N4z, W2.
> ```
> Q1: Prediction on ShareGPT
> ```
> To examine cross-distribution generalization, we report predictor performance on ShareGPT, which is the prompt source used in our scheduling experiments. **For BurstGPT**, only the arrival process is modified, while prompts are still sampled from ShareGPT.
> |Dataset|MAE↓|ACC-50↑|ACC-100↑|
> |-|-|-|-|
> |Alpaca|56.6|58.4%|84.7%|
> |ShareGPT|70.7|46.1%|75.7%|
>
> The predictor maintains reasonable accuracy under this distribution shift, with only moderate degradation, which sufficient to support downstream scheduling decisions. Importantly, MAPS incorporates uncertainty-aware calibration, which refines prediction errors online and provides calibrated upper bounds for scheduling.
>
> We also evaluate on additional datasets (please see Review NrUJ, W2&Q2), further supporting generalization.
> ```
> Q2: Routing Objective Sensitivity
> ```
> We implement the earliest available start time (EAST) objective as suggested and evaluate it under high-concurrency settings. The results are shown below.
>
> LLaMA3.1-8B on ShareGPT
> |Methods|Request_rate|E2E P99(s)|E2E Mean(s)|
> |-|-|-|-|
> |MAPS|17.5|**38.609**|**17.618**|
> ||20|**42.761**|**21.923**|
> |EAST|17.5|40.597|17.679|
> ||20|45.402|24.982|
>
> Qwen2.5-3B on BurstGPT
> |Methods|Temporal_scale|E2E P99(s)|E2E Mean(s)|
> |-|-|-|-|
> |MAPS|200|**11.590**|2.992|
> ||205|**11.235**|3.006|
> |EAST|200|12.876|**2.905**|
> ||205|12.601|**2.759**|
>
> The results show that MAPS achieves consistently strong performance across different routing objectives. Using EAST as an alternative still yields competitive results, indicating that MAPS is not highly sensitive to the specific routing policy.
>
> However, MinQueue consistently achieves the best performance, especially in mean latency. Overall, while EAST is a reasonable alternative, MinQueue provides the most stable performance in practice.
> ```
> Q3: Prediction–Prefill Overlap
> ```
> We clarify that SP consists of a single prefill pass without decoding, so its latency depends only on the input prompt and is independent of the output length.
>
> Prediction runs locally without network overhead or multi-request contention, while cloud-side prefill operates under shared load.
>
> For long prompts, although prediction latency increases with input length, prefill latency grows more significantly and remains the dominant cost; in our measurements, **no** requests with >1000 input tokens exceed prefill latency. This indicates that the fraction does not increase under long-prompt workloads.
>
> For short prompts(<100 tokens), prediction is very fast due to its single-pass design, with only **0.8%** of cases exceeding prefill latency.
>
> Importantly, in all cases no timeout fallback is triggered with β = 1s. Overall, prediction is effectively overlapped with prefill, and timeout fallback is rarely activated.
> ```
> Q4: RSC Network Overhead
> ```
> RSC aggregates only lightweight metadata from each decoder (e.g., available KV blocks, queue length, and prediction summaries), typically <100 bytes per update after serialization.
>
> Assuming a 100 ms update interval and 100 decoders, the total traffic is: 100 bytes × 10 updates/s × 100 decoders ≈ 100 KB/s, which is negligible compared to typical intra-cluster bandwidth (≥1 Gbps ≈ 125 MB/s).
>
> In addition, RSC uses Redis only for lightweight key-value updates rather than transferring request contents or KV cache data. This type of periodic state collection is common in distributed cluster management frameworks such as Kubernetes, which maintain significantly richer control-plane state without making bandwidth the bottleneck.
>
> Therefore, the network overhead of RSC is negligible and does not limit scalability.
>
> ---
> We sincerely thank the reviewer for recognizing our contributions. We hope our clarifications address your concerns and would greatly appreciate an increased rating.

---

> > ### Author Rebuttal · Reviewer_PTKc · 2026-03-31
> >
> > My concerns have been fully addressed. I will wait for the other reviewers' feedback and will consider adjusting my score afterward.

---

> > > ### Author Response · Authors · 2026-04-02
> > >
> > > We are very glad that our responses have addressed your concerns. Thank you again for your thoughtful feedback, which has improved this work—we truly appreciate it and look forward to your final assessment.

---

### Official Review · Reviewer_v5hX · 2026-03-09

**Soundness:** 3
**Presentation:** 3
**Significance:** 3
**Originality:** 3
**Overall Recommendation:** 4
**Confidence:** 4

**Summary:**

This paper proposes MAPS, a memory-aware predictive scheduling framework for prefill-decode (PD) disaggregated LLM serving. The core insight is that output length heterogeneity causes persistent KV cache imbalance across decoders under naive round-robin routing, leading to tail latency inflation. MAPS addresses this via three components:
•	A device-side speculative predictor (LoRA-tuned LLM) that runs in parallel with cloud prefill, incurring near-zero latency overhead.
•	An uncertainty-aware calibration module (conformal prediction) that converts point estimates into coverage-guaranteed upper bounds, guarding against dangerous underestimation.
•	A hierarchical global-local scheduler that routes requests to memory-feasible decoders with the shortest queues (global), and applies SJF reordering within each decoder (local).
Experiments on ShareGPT and BurstGPT workloads with LLaMA-3.1-8B and Qwen2.5-3B show 42.6% average E2E latency reduction and up to 84.8% tail latency reduction versus vLLM, SGLang, and Llumnix.

**Compliance With Llm Reviewing Policy:**

Affirmed.

**Key Questions For Authors:**

1. The key latency benefit depends on prediction completing before prefill. What fraction of requests actually hit the timeout fallback in your experiments? Does this fraction grow significantly under long-prompt workloads where prefill is fast but output length prediction is short?
2. In the global scheduler, memory reservations are held atomically to prevent race conditions. Under high concurrency with many simultaneous scheduling decisions, does this reservation mechanism become a throughput bottleneck?
3. SJF reordering is applied within each decoder's ready queue. How does MAPS handle requests whose actual output length significantly exceeds \hat{L}^{up} despite UAC? Will the perform degrade significantly?
4. The speculative predictor is trained to predict the maximum output length across multiple models. Does this result in systematic over-prediction for simpler queries, leading to unnecessary memory reservation and reduced effective throughput?
5. How close is MAPS to the oracle upper bound? The paper benchmarks against reactive baselines (vLLM, SGLang, Llumnix), but never compares against an oracle scheduler that has access to the ground-truth output length of each request at dispatch time.

**Limitations:**

yes.

**Strengths And Weaknesses:**

Strength
1. Well-motivated problem. The failure of round-robin under heavy-tailed output distributions is clearly demonstrated
2. Applying conformal prediction to bound underestimation risk is a meaningful contribution. Unlike ad-hoc offset heuristics, UAC provides finite-sample statistical guarantees and adapts to distribution shift via a sliding calibration window.
3. The observation that "minimum queue" outperforms "maximum capacity" routing (MQ→MC ablation) is non-obvious and practically important.

Weakness
1. Limited scalibility: All experiments are conducted on 6 GPUs. The scheduling benefits of MAPS are most pronounced under heterogeneous, multi-decoder deployments. It is unclear whether the global scheduler's policy scales well to hundreds of decoders common in large-scale inference clusters
2. The UAC sliding window requires 500 completed requests to calibrate. Coverage guarantees are invalid during the initial warm-up period.
3. The speculative predictor is trained on outputs from LLaMA-3.1-8B and Qwen-2.5-3B. Generalization to new downstream models requires retraining. This maintenance overhead limits practical deployment flexibility.
4. The prediction depends on the predefined timeout threshold, and MAPS assumes every user request originates from a device capable of running a 7B-parameter LLM predictor. This is a strong constraint.

---

> ### Author Rebuttal · Authors · 2026-03-31
>
> We thank the reviewer for the recognition and address the concerns below with additional evidence on MAPS.
> ```
> W1: On scalability
> ```
> Please see Reviewer 2N4z, W2.
> ```
> W2: Warm-up phase
> ```
> (1) **Strong performance during warm-up.**  MAPS remains fully functional before UAC activation, relying on SP with core scheduling (MinQueue + SJF) already in effect. Under ShareGPT at 15 req/s, MAPS outperforms baselines during warm-up (e.g., **41.61s** P99 vs. 57.16s/56.53s for vLLM/SGLang), indicating that UAC is a refinement rather than the primary source of gains. For clarity, see details：https://anonymous.4open.science/r/Maps-09B3/v5hX/W2.md.
>
> (2) **Short period.** The calibration window is quickly populated; e.g., at 15 req/s, 500 requests complete within 79.84s, after which UAC becomes active.
>
> (3) **Flexible window size.** The choice of n=500 is a practical default rather than a strict requirement. Smaller windows also provide effective calibration, with slightly looser bounds.
> ```
> W4: Small scale of SP
> ```
> Please see Reviewer 2N4z, W1.
> ```
> Q1：Timeout Fraction
> ```
> Please see Reviewer PTKc, Q3.
> ```
> Q2：Reservation Overhead
> ```
> The reservation mechanism does not become a throughput bottleneck.
>
> (1) Negligible overhead. Reservation is a lightweight metadata operation (<1 ms in our measurements), which is orders of magnitude smaller than decode latency (~20-70s), and thus negligible.
>
> (2) Robust under concurrency. As shown in Fig. 12, even as the request rate increases (100→500 req/s), the scheduling overhead remains stable (~**1ms**) without noticeable growth, and the overall decode latency remains well-controlled.
>
> Importantly, reservation process does not block the scheduling pipeline. Under high concurrency, the scheduler maintains progress by dispatching requests to other feasible decoders, and in near-saturation cases falls back to the decoder with the largest available memory, ensuring continuous progress.
> ```
> Q3: When length exceeds \hat{L}^{up}
> ```
> Performance degrades gracefully rather than catastrophically.
>
> (1) **Bounded probability.** UAC ensures $P(L^{out} \le \hat{L}^{up}) \ge 1-\alpha = 90\%$ (Eq.5). Empirically, underestimation is only 9% (Table 2), with much smaller errors than the uncalibrated predictor.
>
> (2) **Runtime safety net.** If $L^{out} > \hat{L}^{up}$, decoding proceeds normally via vLLM. KV cache is allocated on demand (PagedAttention), and any memory exhaustion triggers standard preemption, identical to baseline behavior.
>
> (3) **No cascading impact.** Under continuous batching, such cases do not block others. The affected request may run longer, but others continue concurrently; in the worst case, its behavior degrades to baseline-level, while others still benefit from MAPS.
>
> This is validated by the w/o UAC ablation (Fig. 5): even with 64% underestimation, MAPS still outperforms all baselines, demonstrating strong robustness.
> ```
> Q4: Impact of Overestimation
> ```
> Overestimation does not lead to harmful system-level effects in practice.
>
> (1) **Mild rather than systematic.** On ShareGPT, for requests with output length <100 tokens, 61.4% are overestimated, which is only slightly above the probability of overestimation under a symmetric error assumption.
>
> (2) **No actual memory waste.** Overestimation does not incur real memory waste, as predicted length is used only for scheduling, while KV cache allocation grows with the actual generated tokens (PagedAttention).
>
> (3) **Limited system impact.** While overestimation may lead to slightly more conservative admission, reservations are lightweight and quickly released.
>
> Overall, this reflects a deliberate trade-off: mild overestimation is tolerated to avoid underestimation, which can cause resource contention and severe delays.
> ```
> Q5: Comparison to Oracle
> ```
> We construct an Oracle scheduler using ground-truth output lengths and compare its performance with MAPS.
>
> MAPS closely tracks Oracle in tail performance: E2E P99 remains within a modest gap (~5–7 s across load levels), corresponding to a relative difference of ~19.8%, which further narrows under high concurrency. For clarity, see: https://anonymous.4open.science/r/Maps-09B3/v5hX/Q5.md.
>
> **Interestingly**, MAPS achieves slightly lower mean latency than Oracle. This may be attributed to the oracle’s exact allocation with perfect memory fitting, which can restrict long requests to a limited set of feasible decoders and potentially affect load balance. In contrast, imperfect prediction may allow long requests to be dispatched more broadly, albeit with the cost of increased contention, which affects tail latency.
>
> Overall, MAPS achieves near-oracle tail performance while often outperforming Oracle in mean latency, confirming that imperfect prediction (including the cases discussed in Q3) does not significantly degrade performance.
>
> ---
> We hope these responses address your concerns and would appreciate a more favorable score.

---

> > ### Author Rebuttal · Reviewer_v5hX · 2026-04-03
> >
> > Thanks for clarifying. I will maintain my score.

---

> > > ### Author Response · Authors · 2026-04-04
> > >
> > > We are glad that our responses have addressed the concerns. We would greatly appreciate it if the reviewer could reconsider the score in light of these clarifications.

---

### Official Review · Reviewer_2N4z · 2026-03-12

**Soundness:** 2
**Presentation:** 3
**Significance:** 3
**Originality:** 3
**Overall Recommendation:** 4
**Confidence:** 4

**Summary:**

This paper proposes MAPS, a framework that addresses the scheduling challenges of LLM workloads with unknown response length. It designs a length-aware scheduling based on calibrated response lengths estimated by a fine-tuned LLM predictor. The authors conduct studies on the effect of each scheduling module and comparison experiments with popular LLM serving baselines. The results demonstrate better performance of MAPS on both efficiency and memory feasibility.

**Compliance With Llm Reviewing Policy:**

Affirmed.

**Final Justification:**

The rebuttal addresses most of my questions, especially by showing that MAPS can still work with a much smaller 160M predictor and by clarifying the rationale behind SP+Offset versus SP+UAC. However, my concerns about scalability and applicability to CoT/reasoning models are only partially resolved, since these points are argued mainly conceptually rather than supported by additional experiments.

Therefore, I would like to keep my original score.

**Key Questions For Authors:**

1.	As the paradigm of Large Reasoning Models has emerged, the lengths of their chain-of-thoughts are largely extended and varied. When MAPS is transferred to long-context or reasoning model serving, can the prediction on response lengths still be reliable enough to improve the efficiency of scheduling?

2.	The training details mention that the speculative predictor is a 7B model. It still requires a corresponding scale of computational resources on each request-origin device instead of the cloud-side server. Is there any experiment on the minimum scale of the predictor that can support the scheduling?

3.	In the SP+Offset setting in Section 4.1, why is the offset ratio chosen as 10%?  Moreover, if the ratio is set larger such that its underestimation probability decreases and is comparable to SP+UAC, will the scheduling framework that replaces SP+UAC with SP+Offset also exhibits comparable performance?

**Limitations:**

The limitation is not discussed. MAPS have  to be evaluated on larger-scale serving systems and Large Reasoning Models, with lightweight LLM predictor model.

**Strengths And Weaknesses:**

Strengths:
S1: The paper leverages the predictability of LLM response length to solve an important problem of LLM serving scheduling.

S2: The evaluation metrics and workload settings are comprehensive and well-designed, which provide insights into performance improvement of MAPS.

Weaknesses:
W1: A potential weakness is the practicality of the device-side “lightweight” predictor. The paper assumes the predictor can run on request-origin devices with negligible overhead, yet the implementation uses Vicuna-7B, which may still be too heavy for many realistic end devices. This makes the deployment assumption somewhat strong and weakens the claim that the predictor is truly lightweight, especially since one of the main served models is Qwen2.5-3B and LLaMA-3.1-8B.

W2: The scale of the serving system and the LLM for workload in the experiment is limited. Such settings might not  fully the scheduling of large-scale model serving systems.

W3: The paper mainly focuses on basic chat LLMs such as LLAMA-3.1-8B, while not evaluating MAPS on models with enhanced Chain-of-Thoughts reasoning capability.

---

> ### Author Rebuttal · Authors · 2026-03-31
>
> We sincerely thank you for the constructive feedback and for recognizing the role of speculative prediction. We address the main concerns below.
> ```
> W1&Q2: Small scale of SP
> ```
> We clarify that the 7B speculative predictor is used as a reference implementation, as such models are increasingly feasible on modern edge devices. However, MAPS does not depend on large predictors and can operate effectively with much smaller models that provide reliable prediction signals.
>
> To examine this, we evaluate a much smaller 160M predictor (SP-160M). As expected, it exhibits lower prediction accuracy. However, its predictions remain informative for downstream scheduling decisions, and are further refined by UAC. As a result, MAPS (160M) still achieves strong system performance. Compared to existing systems, it improves E2E P99 latency by **21.5% / 21.0% / 56.8%** and mean latency by **12.9% / 6.6% / 59.1%** over SGLang, vLLM, and Llumnix, respectively. (For clarity, see figure: https://anonymous.4open.science/r/Maps-09B3/2N4z/Q2.md; allowed by rebuttal guidelines)
>
> |Model|MAE↓|ACC-50↑|ACC-100↑|
> |-|-|-|-|
> |SP|56.6|58.4%|84.7%|
> |SP-160M|92.2|44.8%|64.5%|
>
> Therefore, MAPS does not rely on large predictors and remains effective with much smaller models, making it well-suited for deployment on resource-constrained edge devices.
> ```
> W2: On scalability
> ```
> While large-scale deployment is an important direction, our goal is to evaluate the effectiveness of the scheduling mechanism rather than system-scale engineering.
>
> The key challenges we address, memory contention, queueing dynamics, and load imbalance under heterogeneous request lengths, already emerge clearly in our current setting and are sufficient to stress the scheduler.
>
> MAPS is architecturally scalable to large clusters. The global scheduler relies only on lightweight, per-decoder metadata, making each scheduling decision a simple $O(N)$ lookup. As shown in Fig.12, the MAS overhead is merely ~1 ms even under high stress. Because state collection is asynchronous and decoupled from the critical path, adding more decoders does not block request dispatching.
>
> The resulting overhead is therefore expected to grow only **mildly** with the number of decoders.
> ```
> W3&Q1: On CoT models
> ```
> MAPS is designed for general LLM serving scenarios, where total generation length is a primary driver of serving cost. In these settings, our results show that memory-aware scheduling already yields substantial improvements in efficiency.
>
> In CoT-based reasoning models, the serving cost is dominated not only by the final output but also by intermediate reasoning tokens, which are dynamically generated and highly variable, as noted by the reviewer. This makes accurate estimation of the full token budget (including reasoning traces) significantly more challenging. To the best of our knowledge, reliable prediction of CoT reasoning length remains an open problem, as existing work primarily focuses on controlling or compressing reasoning rather than predicting it.
>
> Importantly, this challenge lies in prediction. The core framework (memory-aware routing + SJF) remains applicable, while only the prediction module requires adaptation. UAC provides conservative safeguards for high-variance workloads, and the modular design of MAPS allows future CoT-aware predictors to be seamlessly integrated.
>
> We believe this is a promising direction and are actively exploring it as future work.
> ```
> Q3: Why 10% offset, and can a larger offset match UAC?
> ```
> No. A fixed offset faces an inherent accuracy–coverage trade-off that UAC avoids through adaptive calibration.
>
> (1) **Why 10%.** The 10% offset follows common heuristic practice and serves as a representative baseline. As shown in Table 2, it only marginally reduces underestimation (64% → 56%), indicating that small fixed offsets are insufficient.
>
> (2) **Why larger offsets cannot replace UAC.** To match UAC’s ~9% underestimation rate, the offset must be substantially increased (≈35–40% based on Appendix B, Fig. 9). However, this leads to two issues:
> - **Heterogeneous errors.** A single global ratio cannot fit all requests: it remains insufficient for hard cases while severely overestimating easy ones.
> - **Over-reservation.** Excessive overestimation leads to larger memory reservations, causing decoders to fill up more quickly. This reduces the number of feasible decoders for incoming requests, increases queueing, and results in higher latency.
>
> In contrast, UAC applies an adaptive correction based on the empirical error distribution, achieving low underestimation without uniform inflation. This explains why SP+UAC achieves much lower underestimation while maintaining strong scheduling performance (Fig. 5).
>
> We will include additional comparisons across offset ratios in the camera-ready to further illustrate this trade-off.
>
> ---
> We hope our responses have addressed your concerns. We would greatly appreciate your consideration in raising our score.

---

> > ### Author Rebuttal · Reviewer_2N4z · 2026-04-02
> >
> > The rebuttal addresses most of my questions, especially by showing that MAPS can still work with a much smaller 160M predictor and by clarifying the rationale behind SP+Offset versus SP+UAC. However, my concerns about scalability and applicability to CoT/reasoning models are only partially resolved, since these points are argued mainly conceptually rather than supported by additional experiments.
> >
> > Therefore, I would like to keep my original score.

---

> > > ### Author Response · Authors · 2026-04-03
> > >
> > > We thank the reviewer for the detailed follow-up comments and for acknowledging our responses to the earlier concerns.
> > >
> > > ```
> > > Additional Experiments on CoT Models
> > > ```
> > > To further address the applicability of MAPS to CoT-style reasoning models, we conduct additional experiments by explicitly modeling the overhead introduced by intermediate reasoning tokens.
> > >
> > > Our framework is inherently compatible with CoT scenarios, as the core scheduling components (memory-aware routing and queue-aware ordering) are agnostic to the source of token consumption. The main challenge lies in prediction: unlike standard LLMs, CoT models generate additional intermediate reasoning tokens that are unobservable and highly variable.
> > >
> > > To account for this, we augment the SP prediction with a constant offset (+200 tokens). The SP is originally designed to estimate the final output length of standard LLM requests. The offset approximates the additional tokens introduced by intermediate reasoning steps. As a result, the predictor estimates an effective token budget that captures both final outputs and reasoning tokens.
> > >
> > > We evaluate MAPS on two reasoning-capable models, DeepSeek-R1-Distill-Qwen-7B and Qwen3-VL-8B-Thinking, under ShareGPT workloads with request rates of 10, 15, and 20 req/s. As shown in Table, MAPS consistently outperforms vLLM across all settings, achieving an average improvement of **20.07%** in P99 E2E latency and **12.97%** in mean E2E latency.
> > >
> > > These results empirically demonstrate that MAPS naturally extends to CoT-style models. With more advanced CoT-aware predictors, we expect further improvements in scheduling efficiency.
> > >
> > > **DeepSeek-R1-Distill-Qwen-7B (ShareGPT)**
> > > |Methods|Request_rate|E2E P99 (s)|E2E Mean (s)|
> > > |-|-|-|-|
> > > |MAPS|10|**20.125**|**5.300**|
> > > ||15|**20.690**|**5.660**|
> > > ||20|**21.853**|**5.905**|
> > > |vLLM|10|20.959|6.000|
> > > ||15|22.334|6.791|
> > > ||20|28.902|9.469|
> > >
> > > **Qwen3-VL-8B-Thinking (ShareGPT)**
> > > |Methods|Request_rate|E2E P99 (s)|E2E Mean (s)|
> > > |---|---|---|---|
> > > |MAPS|10|**23.477**|**7.098**|
> > > ||15|**26.709**|**9.167**|
> > > ||20|**34.789**|**14.748**|
> > > |vLLM|10|31.564|9.068|
> > > ||15|38.470|13.470|
> > > ||20|38.542|15.273|
> > >
> > > ```
> > > Scalability: Empirical Evidence and Design Analysis
> > > ```
> > > While our evaluation is conducted on a 6-GPU cluster, we provide complementary empirical evidence and architectural analysis to support scalability.
> > >
> > > **Empirical evidence under increasing system pressure.**
> > >
> > > Instead of scaling the number of GPUs, we stress the system along key dimensions that directly impact scalability, including request rate (10–20 req/s), bursty arrival patterns, and model heterogeneity. Across all these settings (Fig. 4 and Fig. 12), MAPS consistently outperforms strong baselines, indicating that its effectiveness persists as system load and scheduling complexity increase.
> > >
> > > These conditions emulate key challenges in large-scale deployments, where higher concurrency and more diverse workloads intensify memory contention and queueing effects.
> > >
> > > **Architectural Analysis of Scalability.**
> > >
> > > MAPS is designed with lightweight scheduling:
> > >
> > > 1) Scheduling decisions rely only on per-decoder metadata (queue length and memory state), resulting in constant-time decision complexity.
> > > 2) The runtime state collector operates asynchronously and is decoupled from the critical path.
> > > 3) As shown in Fig.12, the scheduling overhead remains negligible (**~1 ms**), which is orders of magnitude smaller than decoding latency.
> > >
> > > Therefore, increasing the number of decoders does not introduce synchronization bottlenecks, and the overhead is expected to grow sub-linearly.
> > >
> > > **Summary.**
> > >
> > > Taken together, both empirical results under increasing system stress and the lightweight design suggest that MAPS can scale to larger deployments while maintaining its scheduling benefits. Extending the evaluation to larger-scale deployments is part of our ongoing work.

---

### Decision · Program_Chairs · 2026-04-30

**Decision:**

Accept (regular)

**Comment:**

MAPS addresses the important problem of output-length uncertainty in disaggregated LLM serving through a well-designed combination of speculative prediction, uncertainty-aware calibration, and hierarchical scheduling, achieving 42.6% average E2E latency reduction and up to 84.8% tail latency reduction. The review process reached a strong majority positive consensus, and the rebuttal effectively addressed the main concerns by demonstrating that a 160M predictor still yields meaningful gains and that MAPS tracks oracle performance closely. Concerns about scalability to larger models and chain-of-thought workloads remain partially open but are appropriately scoped as future work. We recommend acceptance and ask the authors to include the oracle comparison and lightweight predictor results in the camera-ready version, along with an explicit limitations discussion.